

# An ethogram of facial behaviour in domestic horses: evolutionary perspectives on form and function

Kate Lewis[1], Sebastian D. McBride[2], Jérôme Micheletta[1], Matthew O. Parker[3], Alan V. Rincon[1], Jen Wathan[4] and Leanne Proops[1]

[1] Centre for Comparative and Evolutionary Psychology, University of Portsmouth, Portsmouth, United Kingdom
[2] Institute of Biological, Environmental and Rural Sciences, Aberystwyth University, Aberystwyth, United Kingdom
[3] School of Bioscience, University of Surrey, Surrey, United Kingdom
[4] The Brooke Hospital for Animals, London, United Kingdom

Corresponding author
Kate Lewis, kate.lewis@port.ac.uk

## ABSTRACT

Understanding cause and function of animal facial behaviour can provide key insights into the animal's cognitive and emotional state. The Equine Facial Action Coding System (EquiFACS) has characterised a wide range of equine (*Equus caballus*) facial movements (Action Units (AU) and Descriptors (AD)). However, there is still a lack of systematic documentation of whether and how these AUs and ADs are combined to produce discrete configurations of facial behaviour in horses. The aim of this study was to provide a systematically derived ethogram of equine facial behaviour in the domestic horse across positive, negative and neutral social interactions. Facial behaviour was recorded during horse-horse interactions occurring during affiliative (non-play), play, agonistic and attentional contexts, based on the coding of contextual behaviour. Using EquiFACS, a bank of 805 AU/AD combinations, across 22 distinct behaviours, was created. Network analysis techniques (NetFACS) were used to establish the facial movements significantly associated with each context. Domestic horses use a wide range of distinct facial behaviours, which are defined and described in our ethogram. Interestingly, there were marked similarities between the play faces of horses and the open mouth play faces of primates and carnivores, adding weight to the hypothesis that these facial behaviours are deep rooted in mammalian biology. We also defined a new EquiFACS Action Unit, AUH21, facial tightener (platysma), which makes the underlying facial structures appear more prominent. This AU is found in humans and gibbons, but no other species studied, and its addition to EquiFACS will enhance cross-species comparisons and potentially facilitate the attribution of emotional state and assessment of pain in horses. Our equine facial behaviour ethogram will be invaluable in future work exploring equine emotion, welfare, social behaviour, and perception, as well as having direct applications for those working with and around horses.

# INTRODUCTION

Facial behaviour (observable changes in the appearance of the face) is used extensively in humans, but is also seen throughout the primate order and in other mammals, including horses (*Waller & Micheletta, 2013*). The term "facial expression" tends to imply associated emotion, which is not always the case, hence we use the term "facial behaviour" throughout (*Waller, Julle-Daniere & Micheletta, 2020*). Facial behaviour provides a valuable source of information about the signaller, their future behaviour, and potentially their internal state (*Waller & Micheletta, 2013*). Accurate production and perception of facial behaviour therefore provides an evolutionary advantage for group-living species reliant on the visual modality for communication and the management of individual relationships to ensure group cohesion (*Pusey & Packer, 2003*). In this context, understanding the cause and function of animal facial behaviour provides considerable insight into the animal's cognitive and emotional state, and in predicting future behaviour (*Erickson & Schulkin, 2003*).

In social species, facial behaviour is central to maintaining group cohesion by regulating social interactions. For example, recent studies focusing on the silent bared teeth display revealed that just like in humans (*Martin et al., 2017*), this configuration of the face can be highly variable and context specific in some primate species. In crested macaques (*Macaca nigra*), different variants can facilitate mating, affiliation, play or function as a submissive signal (*Clark et al., 2020*). Similarly, the relaxed-open mouth is often used to punctuate playful interactions in a number of mammalian species including dogs (*Canis lupus familiaris, Maglieri et al., 2022*), wolves (*Canis lupus, Maglieri et al., 2024*), sea lions (*Otaria flavescens*, *Llamazares-Martín et al., 2017*), and horses (*Equus caballus*, *Maglieri et al., 2020*). Experimental evidence also suggests that facial appearance changes can be used by receivers to anticipate the outcome of social interactions, potentially as a way to minimise the risk of costly physical conflicts (*Waller, Whitehouse & Micheletta, 2016*).

While there is a general consensus that animals have rich and potentially complex emotional experiences, the nature of animal emotions and whether they can be reliably inferred from facial behaviour remains a contentious topic. Even in humans, facial behaviour alone is a poor predictor of felt emotion (*Durán & Fernández-Dols, 2021*; *Feldman Barrett et al., 2019*). Nevertheless, there is some evidence that animals can categorise static facial configurations to situations associated with positive or negative emotional valence (*Parr, 2001*) and that facial behaviour can elicit behavioural and physiological changes in others (*Kemp & Kaplan, 2013*; *Kuraoka & Nakamura, 2011*). There are also striking similarities in the facial behaviour of humans and other species (*Kavanagh et al., 2022*; *Kret et al., 2020*), although there are interspecies differences in the facial behaviour displayed within contexts (for examples see *Caeiro, Guo & Mills, 2017*). Facial behaviour therefore remains an important area of research for scientists interested in emotions.

Understanding and interpreting facial behaviour also has important practical implications, notably regarding animal welfare, particularly in domestic animals and those used in sports (*Descovich et al., 2017*). Changes in facial behaviour can be associated

with painful, stressful, or generally uncomfortable situations. For instance, numerous pain assessment scales based on specific facial action units like orbital tightening, nose bulges, and ear positions, have been developed for various species, including rodents (*Mus musculus*: *Langford et al., 2010*; *Rattus norvegicus*: *Sotocinal et al., 2011*), rabbits (*Keating et al., 2012*), and horses (*Dalla Costa et al., 2014*; *Rashid et al., 2020*). Accurate identification and interpretation of these cues can potentially allow caregivers and researchers to improve animal welfare by addressing pain and distress more effectively but also ensure that individuals display a range of natural behaviour (*Descovich et al., 2017*).

In spite of efforts to standardise facial behaviour analysis, facial behaviour has historically been challenging to record and analyse accurately; it is subject to a high degree of observer bias and is often influenced by the perceived emotional context (*Bruce & Young, 2012*; *Hole & Bourne, 2010*; *Waller et al., 2007*). The Facial Action Coding System (FACS) was developed to overcome these limitations and provide a reliable, standardised, and systematic framework for documenting facial actions, based on underlying facial musculature. Originally designed for use in humans (*Ekman & Friesen, 1978*), FACS uses designated codes called Action Units (AUs) corresponding to the contraction of a particular facial muscle, or set of muscles, resulting in specific observable movements. Action Descriptors (ADs) are used to identify more general facial movements, where either the muscular basis is not known or it results from the action of non-facial muscles. Facial behaviour thus comprises a variable number of these AU/ADs, used in combination. FACS has since been successfully applied to a number of different primate and domestic species (*Bennett, Gourkow & Mills, 2017*; *Caeiro et al., 2013*; *Parr et al., 2010*; *Vick et al., 2007*; *Waller et al., 2012*; *Waller et al., 2013*), including horses (*Wathan et al., 2015*). The latter, as a highly social visual species that operates in complex multi-level societies (*Maeda et al., 2021*), requires a correspondingly complex signalling system to allow for effective communication and maintenance of social networks (*Murphy, Hall & Arkins, 2009*). This involves both whole-body postural as well as multifaceted facial movements that are readily identifiable within a FACS analysis system (*Waring, 2003*; *Wathan et al., 2016*). The Equine Facial Action Coding System (EquiFACS) was developed in 2015, yet, although a wide range of equine facial movements have been identified, there is still a lack of systematic and complete documentation of whether and how these movements are combined to produce discrete configurations of facial behaviour in horses. Some authors have previously linked specific facial behaviour in horses with pain and discomfort (*Dalla Costa et al., 2014*; *Dyson et al., 2017*; *Gleerup et al., 2015*; *Rashid et al., 2020*) but characterisation of equine facial behaviour is limited to this particular context (*Dalla Costa et al., 2017*), and to general positive and negative emotional states (*Hintze et al., 2016*; *Lansade et al., 2018*; *Phelipon et al., 2024*; *Ricci-Bonot & Mills, 2023*; *Trösch et al., 2020*). It is critically important, therefore, that we have a broader understanding of the full range of evolved equine facial behaviour that is exhibited during different positive, neutral and negative contexts. This understanding will greatly facilitate future research exploring equine emotion, welfare, and social behaviour.

The aim of this study, therefore, was to provide the first, systematic description of equine facial behaviour using the domestic horse (*Equus caballus*) as an example equid. Facial behaviour was derived from combinations of AUs and ADs during horse-horse

interactions within a range of affiliative and agonistic social contexts, based on their underlying morphology. Note that portions of this article were previously published as part of a pre-print (*Lewis, 2023*).

## MATERIALS & METHODS

### Collection of video footage

Subjects were 36 adult horses (12 mares, 24 geldings) of mixed breeds and aged 5–19 years ($11.3 \pm 3.8$ yrs; mean $\pm$ SD), housed as Sparsholt College Equine Centre, Winchester, UK. This sample size is in line with that of previous facial expression work using network analysis techniques (*Rincon et al., 2023*). Recordings were made between September and December 2020, between 10:00 and 16:00, on days of routine turnout (horses placed out in a large pasture, with freedom to move around). Management of the animals did not alter for the study. During college term-time they were stabled individually between Monday morning and Friday evening, and turned out over weekends (Fri–Mon). During college holidays, they were turned out full time. Data collection took place during these turn-out periods. Date were collected over 24 days, and the total observation time was 72 h. The number of hours of recording on a given day ranged between two and five hours. Horses were turned out in two established same-sex groups; mares and geldings. The field into which each group was turned out changed periodically throughout the study. Group sizes and composition varied between recording sessions, due to changes in horse availability. The group (mares or geldings) recorded on a given day was pseudo-randomly selected, with consideration given to the number of horses within each group. Because of the temporal variation in group sizes and individuals present over the data collection period, we were not able to pre-determine the observation schedule. Initially groups were selected randomly, using a random number generator, and then adjustments were made during later randomisation in order to balance the number of observations of each individual as much as possible.

Two researchers conducted 30 min focal samples, and subject order was pseudo-randomly selected; observations were balanced across the season and time of day as much as possible, given horse availability varied throughout the study. During a focal observation, the researcher stood between approximately 10–20 m from the focal individual (the exact distance varied depending on the location of other individuals and environmental features, and any movement of animals during recording), and kept a Panasonic HC-VXF1 camcorder, mounted on a tripod, with pre-record function enabled, trained on the focal animal. The camera was used in HD mode in order to generate high quality footage for FACS analysis, and the zoom function enabled the researcher to keep the camera trained specifically on the focal animal and their close neighbours. Upon a social interaction between the focal horse and one or more conspecifics, the experimenter began recording. A social interaction was defined as when horses moved to within one horse length of each other and/or when an animal made a noticeable signal towards another individual. During recording, the researcher aimed to keep the face of the focal animal in shot. Recording continued until the interaction between the animals was complete. This was defined as

either when the horses withdrew (moved to > one horse length from each other) or there was a period of social inactivity (horses made no noticeable signals towards one another) of > 3s. In cases of prolonged interaction, it was sometimes necessary to stop and restart recording mid-interaction to prevent video files becoming too large. This was done on an *ad hoc* basis (typically if an interaction lasted > 60s), and recording was re-started immediately. A GoPro Hero 5 (wide-angled; GoPro, San Mateo, CA, USA), attached to the tripod arms, provided secondary coverage of behaviour for coding behavioural context. Opportunistic recording of interactions between non-focal animals were conducted during periods of focal animal inactivity.

### Behavioural context analysis

Videos were initially viewed using Windows Media Player (Microsoft®, Bellingham, WA, USA). The behavioural context of each interaction was determined using a predefined social ethogram (Table 1). This ethogram represents an amalgamation of three widely used equine social ethograms (*Christensen et al., 2002*; *McDonnell & Haviland, 1995*; *McDonnell & Poulin, 2002*), so as to cover the whole range of potential social interactions, encompassing affiliative (non-play), play, agonistic and attentional contexts. Affiliative behaviours are friendly, peaceful interactions between individuals (*del Toro & Nekaris, 2019*). Play involves activities which generate a sense of pleasure and elements of surprise, which appear to have no immediate function (*McDonnell & Poulin, 2002*). Agonistic behavior encompasses aggression, threat, appeasement, and avoidance behavior between conspecifics (*McDonnell & Haviland, 1995*). Attentional contexts were defined here as the purposeful orientation of two or more sense organs towards a stimulus to garner information. For each interaction, we noted: (1) the identity of the subjects involved, (2) the behaviour of each subject according to our established ethogram, (3) the time (s) of any facial behaviour produced by any of the individuals performing a social behaviour. If the face of an individual was not visible (either out of sight or unclear) for most or all of an interaction (approx. > 50%), their data were discarded as facial behaviour analysis would not be possible. Where multiple behaviours occurred in quick succession, *e.g.*, during a play bout, each of these was coded separately. In situations where behaviours from multiple contexts were present, these were coded as that of the behaviour most associated with the face. For example if rear and nip occurred together, this was coded as nip. To ensure consistency, 10% ($n = 81$) of behaviours were coded a second time by the same observer (>2 years later). Intra-observer test-retest reliability was calculated using Cohen's kappa and indicated substantial agreement between coding at the two time points, $\kappa = .74$, $p < .001$. Raw data are available at https://osf.io/zmsvx/.

### Analysis of facial behaviour using EquiFACS

The above process identified 1,326 facial behaviours potentially suitable for coding using EquiFACS; however, some behavioural contexts for some horses were over-represented. To prevent the analyses being biased by these contexts and individuals, a maximum of five facial behaviours for an individual horse for a given behavioural context were selected. These behavioural contexts were balanced for date, time of day, and conspecific identity

![PeerJ]

**Table 1   Ethogram of equine social behaviour.** Adapted from *Christensen et al. (2002)*, *McDonnell & Haviland (1995)* and *McDonnell & Poulin (2002)*. Although initially included, vocalisations were removed from the ethogram as it was often not possible to identify their origin. Sexual (play and non-play) and herding behaviours were also removed as they were rarely or never observed in either group.

| Interaction context | Sub-group | Behavioural context | Description |
|---|---|---|---|
| Affiliative (non-play) | | Follow | Moving along the path of another animal, usually at the same gait (*McDonnell & Haviland, 1995*). |
| | | Contact (friendly) | Affiliative direct contact between individuals, not part of play or grooming. |
| | | Groom initiated | Coat care of another individual; partners stand beside one another, usually head-to-shoulder or head-to-tail, grooming each other's neck, mane, rump, or tail by gently nipping, nuzzling, or rubbing (*Christensen et al., 2002*; *McDonnell & Haviland, 1995*). |
| Agonistic | | Displace | Approach of one horse, causes another to move away so that distance is maintained or increased, without overt aggression (*Christensen et al., 2002*). |
| | | Head threat | Horse threatens another with ears back. May be accompanied by strong tail swishing, arched neck, or leg stamping (*McDonnell & Haviland, 1995*). |
| | | Bite threat | Bite intention movement with ears back and neck extended, with no actual contact (*Christensen et al., 2002*; *McDonnell & Haviland, 1995*). |
| | | Kick threat | Kick intention movement, performed by swinging rump or backing up, or by waving or stamping hindleg toward another horse, without making contact (*Christensen et al., 2002*; *McDonnell & Haviland, 1995*). |
| | | Bite | Opening and rapid closing of the jaws, with actual contact to another horse's body. Ears are back and lips retracted (*Christensen et al., 2002*; *McDonnell & Haviland, 1995*). |
| | | Kick | One or both hindlegs lift off the ground and rapidly extend backwards toward another horse, with apparent intent to make contact (*Christensen et al., 2002*; *McDonnell & Haviland, 1995*). |
| | | Strike | One or both forelegs are rapidly extended forward as though to contact another animal (although contact may not be made), while the hind legs remain on the ground. May also occur during rearing (*McDonnell & Haviland, 1995*). |
| | | Chase | One horse chases another (trotting or galloping), with ears laid back (*McDonnell & Haviland, 1995*). |
| | | Push | Pressing of the head, neck, shoulder, chest, or body against another horse, causing it to move one or more legs to regain balance (*Christensen et al., 2002*; *McDonnell & Haviland, 1995*). |
| Attentional | | Alert | Rigid stance with the neck elevated and the head oriented toward the object or animal of focus. Ears are held stiffly upright and forward. Nostrils may be dilated (*McDonnell & Haviland, 1995*). |
| | | Olfactory investigation | Sniffing various parts of another individual's head and/or body, typically beginning after the horses have approached one another nose to nose (*Christensen et al., 2002*; *McDonnell & Haviland, 1995*). |

**Table 1** (*continued*)

| Interaction context | Sub-group | Behavioural context | Description |
|---|---|---|---|
| Play | | Frolic | Fore- and hindlegs simultaneously propel off the ground. Usually accompanied by random bucking, head shaking and body twists (*McDonnell & Poulin, 2002*). |
| | | Leap | As frolic but the body is propelled over, away from, or towards an object (*McDonnell & Poulin, 2002*). |
| | | Run | Seemingly spontaneous movement, at a trot, canter or gallop, with no apparent destination or threat to escape (*McDonnell & Poulin, 2002*). |
| | | Chase (play) | Animal pursues another, at trot, canter or gallop, in order to catch up and overtake it (*McDonnell & Poulin, 2002*). |
| | Locomotor play | Buck | With head and neck lowered and weight shifted to the forelegs, both hindlegs are lifted off the ground and simultaneously extended backwards. May be repeated in quick succession (*McDonnell & Poulin, 2002*). |
| | | Prance | Walk or trot with the neck arched, ears forward, tail elevated and exaggerated knee action (*McDonnell & Poulin, 2002*). |
| | | Nip air | During a play bout, jaws and teeth are opened and closed in the vicinity of the flesh (or rug) of another animal, but without teeth contacting the flesh (or rug). Lips or face may make contact (personal observation). |
| | | Nip | During a play bout, jaws and teeth are opened slightly, closed and quickly released on a small piece of hair or flesh of another animal (or their rug if wearing one) (*McDonnell & Poulin, 2002*). |
| | | Bite (play) | During a play bout, jaws and teeth are opened widely, closed and quickly released on a large piece of skin/flesh of another. Lips may be retracted (*McDonnell & Poulin, 2002*). |
| | | Reach | Horse moves its head towards another individual, as if going to make contact, however there is no contact made. Lips and teeth may be parted, but there is no rapid closing of the mouth as there is in nip air (personal observation). |
| | | Grasp | Jaws and opened and clamped around the flesh of another individual. There may be some movement of the flesh back and forth (*McDonnell & Poulin, 2002*). |
| | | Neck wrestle | Horses spar with their heads and necks. Can be performed standing, on the knees, or with raised forelegs (*McDonnell & Poulin, 2002*). |
| | | Push (play) | During a play bout, head, neck, shoulder, chest, body or rump is pressed against another, in an apparent attempt to displace them ((*McDonnell & Poulin, 2002*). |
| | | Stamp | One foreleg is raised and lowered, striking the ground sharply and firmly (*McDonnell & Poulin, 2002*). |
| | | Rear | Forequarters are raised off the ground whilst the hindlegs remain on the ground, resulting in a near-vertical position (*McDonnell & Poulin, 2002*). |
| | Play fight | Kick threat (play) | During a play bout, rump is turned towards a conspecific and one or both legs are raised as if aiming to kick. Often accompanied by backing up towards the target (*McDonnell & Poulin, 2002*). |
| | | Kick (play) | During a play bout, one or both hindlegs are lifted off the ground and extended backwards towards another animal, without sufficient force to touch or cause injury (*McDonnell & Poulin, 2002*). |
| | | Balk (play) | During a play bout, abrupt reversal of direction of the forebody, withdrawing the head and neck in a sweeping dorsolateral motion whilst the hindbody remains in place or pivots. Forelegs may simultaneously lift off the ground (*McDonnell & Poulin, 2002*). |

**Table 1** (*continued*)

| Interaction context | Sub-group | Behavioural context | Description |
|---|---|---|---|
| | | Evasive jump | Contact is avoided by propelling the fore-, hind- or entire body off the ground, away from the gesture of another (*McDonnell & Poulin, 2002*). |
| | Play evasion | Evasive spin | Contact is avoided by pivoting around one hindleg in a quick, sharp motion (*McDonnell & Poulin, 2002*). |
| | | Head snatch | Head is rapidly moved away from another horse, without movement of the rest of the body. May be accompanied by a squeal (personal observation). |

**Notes.**
There is some overlap between behaviours seen in play and agonistic contexts, and these may be difficult to distinguish in isolation. Here, we categorised such overlap behaviours as play if they were part of a larger play bout, where both horses were engaged and the level of overt aggression was low. Agonistic contexts involved more overt aggression and in this population were short lived, with one individual typically moving away as the result of aggression from another.

wherever possible, with the aim to avoid clustering of examples within a relatively short time frame or with the same social partner. Facial behaviours were randomly selected where there were multiple examples from an individual within one focal sample or on one day. For example, if, for horse one, 12 occasions of the behaviour 'displace' had been identified, five instances of these would be pseudo-randomly selected for FACS analysis. This threshold of five was chosen to reduce over-representation whilst still allowing for within-context individual variation in facial behaviour to be captured. Following this process, 805 facial behaviours remained for EquiFACS coding.

A certified EquiFACS coder (KL) conducted video coding using open source software BORIS (*Friard & Gamba, 2016*). For each facial behaviour, the peak of the facial behaviour during each behavioural context was identified, by selecting the point where the greatest number AU/ADs were activated and appeared to be at their greatest intensity. In cases where the facial behaviour was held static for any length of time, the middle of this period was selected as the peak. Analyses were conducted over a 1s period surrounding the peak (0.5s either side). The peak of the facial behavior always fell within the performance of the behavioural context, however in some cases the upper and/or lower bounds of the 1s time bracket would fall outside of the behavioural context, particularly when behaviours were short-lived. Coding over a 1s window allowed for the dynamic nature of facial behaviour to be captured, which using a still image does not. We appreciate that not coding for the entirety of a behaviour may have resulted in some AU/ADs being missed, however FACS coding is extremely time intensive and, as such, 1s represents a compromise between precision and time efficiency. Videos were viewed at both full speed and at 0.1x speed. AUs and ADs were coded as having occurred (1) or not having occurred (0) during the 1s period. If an AU/AD was not visible at any point during the clip—due to head orientation, poor focus, or blurring due to the speed of movement—and its use could not be confirmed when it was visible or from resulting facial movements, it was marked as out of sight. Videos were compared with images of the focal horses' neutral face, wherever this was available, to allow for more accurate coding. Neutral images, *i.e.,* when horses were relaxed and demonstrating no discernible facial behaviour, were obtained during data collection by opportunistically recording horses during periods of inactivity. AUs and ADs contained within the EquiFACS manual (*Wathan et al., 2015*), as well as the supplementary behaviour and head movement ADs described in the supplementary material, were initially coded (for
a full list see Table 2). AD50 (vocalisation) was later excluded as it was not usually possible to identify the specific animal a vocalisation originated from. Both AD133 (blow) and AD38 (nostril dilator) were coded as AD38 as they were often indistinguishable. To ensure accurate coding, the coding was completed by a certified EquiFACS coder and verified by an independent EquiFACS certified coder who coded 5% ($n = 60$) of a larger bank of videos ($n = 1181$), of which the data used here are a subset (33 of these 60 were videos used in the present study). Inter-coder reliability was determined using the following equation, as recommended for use in human FACS (*Ekman & Friesen, 1978*):

$$\frac{2\left(\#AU \ and \ ADs \ agreed \ by \ both \ coders\right)}{\left(\#AU \ and \ ADs \ coded \ by \ coder \ 1\right) + \left(\#AU \ and \ ADs \ coded \ by \ coder \ 2\right)}.$$

This calculation gives an agreement for each facial behaviour between 0 and 1 (0 = no agreement, and 1 = absolute agreement). Agreement was 0.98, which is well above the criteria required to become a certified FACS coder (0.70).

## Statistical analysis

Network analysis techniques, designed for the analysis of FACS data (NetFACS), were used to analyse the data in R v.4.2.3 (*R Core Team, 2023*) using the package NetFACS *v. 0.5.0.9001* (*Mielke et al., 2021*). Any AU/ADs used less than five times across the whole dataset ($n = 8$, Table 2) were removed, as rare actions create a high degree of uncertainty in the bootstrapping process when using NetFACS (*Mielke et al., 2021*). Data were initially categorised into four broad interaction contexts; affiliative (non-play), play, agonistic, and attentional. Network analyses were performed, using the multi.facs function, on each interaction context to establish the AU/ADs which occurred significantly ($P < 0.01$) more often than observed across all other contexts. AU/ADs were considered 'frequently used' if they occurred in $\geq 40\%$ of facial behaviours within a given context. AU/ADs that occurred in $< 10\%$ of the data were omitted from results, as no meaningful interpretation was possible. Bipartite networks, that show the paired combination of AU/ADS that are co-activated significantly more than chance, were plotted for each interaction context using the netfacs_network function and modified using the ggraph v.2.1.0 package (*Pedersen, 2022*). In co-activation networks, connections between AU/ADs were significant if they occurred at a higher probability in the selected behavioural context than would be expected of that combination across all other contexts. Thus it is possible for an AU/AD combination to be significant whilst the individual AU/ADs that make it up are not, as the baselines for calculating significance differ.

The specificity with which AU/AD was associated with an interaction context was calculated using the specificity function. Due to an imbalanced number of observations across interaction contexts, contexts with fewer observations were randomly upsampled prior to the specificity calculation, to correct for any bias in the specificity results from an imbalanced dataset (*Rincon et al., 2023*). Specificity is the conditional probability of an interaction context given that an AU/AD is observed. It ranges from 0 (when an AU/AD is never observed in a context) to 1 (when an AU/AD is only observed in one context). Low specificity values indicate that an AU/AD was used flexibly across multiple contexts,

**Table 2  EquiFACS action units and action descriptors coded and included in analyses.**

| Facial region/ behavioural category | AU/AD code | AU/AD name | Included in analyses? | Reason for exclusion |
|---|---|---|---|---|
| Eyes | AU101 | Inner brow raiser | Y | |
| | AU143 | Eye closure | N | Observed on <5 occasions |
| | AU145 | Blink | Y | |
| | AU47 | Half blink | Y | |
| | AU5 | Upper lid raiser | Y | |
| | AD1 | Eye white increase | Y | |
| Ears | EAD101L | Left ear forward | Y | |
| | EAD101R | Right ear forward | Y | |
| | EAD102L | Left ear adductor | Y | |
| | EAD102R | Right ear adductor | Y | |
| | EAD103L | Left ear flattener | Y | |
| | EAD103R | Right ear flattener | Y | |
| | EAD104L | Left ear rotator | Y | |
| | EAD104R | Right ear rotator | Y | |
| Lower face | AU10 | Upper lip raiser | Y | |
| | AU12 | Lip corner puller | Y | |
| | AU113 | Sharp lip puller | Y | |
| | AUH13 | Nostril lift | Y | |
| | AU16 | Lower lip depressor | Y | |
| | AU17 | Chin raiser | Y | |
| | AU18 | Lip pucker | Y | |
| | AU122 | Upper lip curl | Y | |
| | AU24 | Lip pressor | Y | |
| | AU25 | Lips part | Y | |
| | AU26 | Jaw drop | Y | |
| | AU27 | Mouth stretch | Y | |
| | AD160 | Lower lip relax | Y | |
| | AD19 | Tongue show | Y | |
| | AD29 | Jaw thrust | N | Observed on <5 occasions |
| | AD30 | Jaw sideways | Y | |
| | AD133 | Blow | N | Combined with AD38 as not possible to distinguish between the two |
| | AD38 | Nostril dilator | Y | |
| Head movements | AD51 | Head turn left | Y | |
| | AD52 | Head turn right | Y | |
| | AD53 | Head up | Y | |
| | AD54 | Head down | Y | |
| | AD55 | Head tilt left | Y | |
| | AD56 | Head tilt right | Y | |
| | AD57 | Nose forward | Y | |
| | AD58 | Nose back | Y | |

**Table 2** (*continued*)

| Facial region/ behavioural category | AU/AD code | AU/AD name | Included in analyses? | Reason for exclusion |
|---|---|---|---|---|
| | AD50 | Vocalisation | N | Not possible to identify individual generating a vocalisation |
| | AD76 | Yawning | N | Observed on <5 occasions |
| | AD80 | Swallow | N | Observed on <5 occasions |
| Gross behaviours | AD81 | Chewing | Y | |
| | AD84 | Head shake side to side | N | Observed on <5 occasions |
| | AD85 | Head nod up and down | N | Observed on <5 occasions |
| | AD86 | Grooming | N | Observed on <5 occasions |
| | AD87 | Ear shake | N | Observed on <5 occasions |

whereas high values indicate that an AU/AD was used primarily in a single context. The probability of occurrence for each AU/AD in each interaction context were extracted from interaction context networks. The probability of occurrence is the conditional probability that a particular AU/AD will be observed in a given context, and ranges from 0 (when an AU/AD never occurs in a context) to 1 (when an AU/AD always occurs in a context). Low values indicate that an AU/AD is rarely used in a particular context, whereas high values indicate that an AU/AD is present in almost all instances of that context. Bipartite networks, showing how single AU/ADs are connected to the four interaction contexts, were plotted for context specificity and probability of occurrence, weighted by AU/AD specificity and probability of occurrence respectively.

Network analyses were also performed on each behavioural context to establish the AU/ADs which occurred significantly ($P < 0.01$) more often in that context than by chance. Bipartite networks were plotted, as described above. The significant AU/ADs and the network plots, in combination with observation of video footage, were utilised to produce written descriptions of facial behaviours typical of each behavioural context, in order to create our facial behaviour ethogram. Images illustrating these typical behaviours were also identified from video footage. R code is available at https://osf.io/zmsvx/.

### Ethics statement

This study was observational and carried out in accordance with the recommendations in the Association for the Study of Animal Behaviour and the Animal Behaviour Society guidelines for the use of animals in research (*ASAB Ethical Committee/ABS Animal Care Committee, 2023*). The study was approved by the University of Portsmouth Animal Welfare and Ethical Review Body (Application No. 1219C).

## RESULTS

### An extension to EquiFACS: AUH21

During analyses, it was noted by researchers that there were changes occurring in the appearance of the equine face during certain interactions (most notably in agonistic and play contexts) that could not be coded using any AU/AD currently available in EquiFACS, yet could play an important role in equine facial behaviour. We thus propose the addition of AUH21 to EquiFACS (Table 3). Personal communication with mammalian facial anatomy

and animal FACS expert Professor Anne Burrows, established that the anatomical origin of the observed appearance change was the action of the platysma muscle. In horses, the platysma is a large superficial muscle of the neck and lower face which primarily functions to shiver off flies. Importantly here, the contraction of this muscle causes tension in the lower face in equines, making the underlying structures appear more prominent. Once defined, all facial behaviour videos were reassessed, using the same method, for the inclusion of AUH21. A subset of videos (30%, $n = 238$) were second coded for AUH21 by an independent certified EquiFACS coder. The agreement across the two coders, calculated using the same formula as described in the methods, was 0.64. This is below the agreement needed to become a certified FACS coder (0.70), however both coders noted that AUH21 is more difficult to code than the other AU/ADs as it is not based on a specific movement of part of the face, rather it is coded when structures appear to become more visible. Visibility can be affected by lighting conditions and the animal's coat (see Table 3 for more information), in a way that discrete movements are not, resulting in an AU which is more difficult to identify than the majority, if not all, of those in the original EquiFACS.

## Overview of interaction contexts

We generated a bank of 805 facial behaviours recorded during conspecific interactions, between 36 domestic horses, in two established social groups. These were categorised into four discrete interaction contexts; affiliative (non-play) ($n = 178$), play ($n = 357$), agonistic ($n = 157$), and attentional ($n = 113$).

The context specificity for single AU/ADs in each interaction context were plotted as a bipartite network, showing how single AU/ADs are connected to the four interaction contexts (Fig. 1). Although there were some AU/ADs that were highly specific to one context, the majority of AU/ADs were used flexibly across multiple different contexts. For clarity, edges are only shown in Fig. 1 where an AU/AD was used in > 10% of occurrences. In reality, almost all AU/ADs were observed in all four contexts at least once. Attentional and play contexts have a number of AU/ADs specifically associated with them. For example, the ears being adducted (EAD102L+R) is highly specific to attentional contexts. In contrast, there are no AU/ADs highly specific to agonistic and affiliative interacts.

The probability of occurrence for single AU/ADs in each interaction context were also plotted as a bipartite network, showing how single AU/ADs are connected to the four interaction contexts (Fig. 2). For each context there were a number of AU/ADs that were highly likely to occur in that context. For example, we can see that lower lip depressor (AU16), lips parted (AU25), ears rotated backwards (EAD104L+R), head down (AD54) and nose forward (AD57) all have high probabilities of occurring during play. There are also many AU/ADs that are relatively rarely used in any context, such as upper lid raiser (AU5) and jaw drop (AU26).

## Affiliative (non-play) interactions

When looking at affiliation overall, one head movement was frequently (*i.e.,* used in ≥40% of interactions) used in affiliative (non-play) contexts; nose forward (AD57) (Fig. 3). The two less frequently used (used in <40% of interactions) face and head

**Table 3  Summary of action unit AUH21 in EquiFACS compared to Human FACS.**

| Action unit | Muscle/s | In human FACS | Appearance changes | Considerations when coding |
|---|---|---|---|---|
| AUH21 Facial tightener | Platysma (a large superficial muscle of the neck and lower face) | Same muscle as AU21, however visual appearance is different due to anatomical differences. | There are no facial movements associated with the AU. Instead, AUH21 is coded if the muscles and structures on the side of the face, between the cheek and muzzle (see region highlighted), become more prominent/visible, or if there is noticeable tightness or tension in this region. These structures may also become more prominent during chewing due to the action of the masseter muscle; if AD81 is present, do not also code AUH21. | AUH21 can be more challenging to identify if the subject: -Has a longer coat and/or is not clipped -Has their mouth open -Is of a stockier breed/is a coldblood type -Has a very dark coat, or the coat is dappled/spotted Lighting should also be taken into consideration when coding AUH21, as this can make the structures of the face appear more or less prominent. Problems with the teeth and/or damage to the inside of the cheek may also cause this region to become more prominent/swollen, and should be considered when coding. |

**Neutral face**

**AUH21 activated**

Note the more prominent muscles (zygomaticus and depressor labii inferioris) and tightness of skin across the marked region.

Note the more prominent muscles (zygomaticus and levator labii superioris) due to the tautness of the overlying skin.

Note the prominent muscles (zygomaticus and depressor labii inferioris), despite the increased length of coat, and the tightness of the skin the marked region.

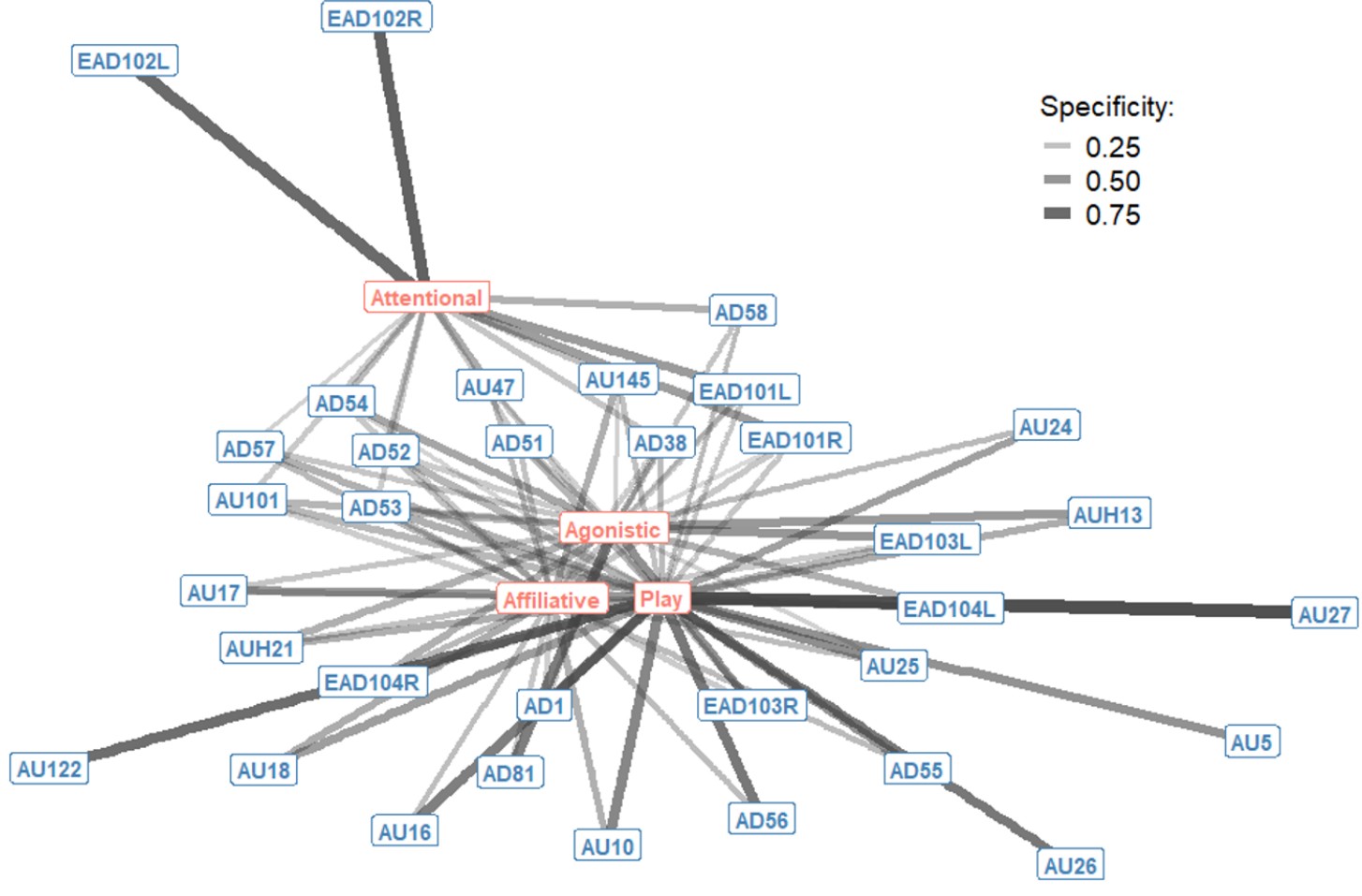

**Figure 1 Context specificity bipartite network of single Action Units and Action Descriptors (blue) and interaction contexts (orange) for domestic horses.** Edges are shown for Action Units and Descriptors that occurred in at least 10% of observations per context. Edge thickness and transparency are weighted by specificity, which ranges from 0 (indicating an Action Unit is never observed in a context) to 1 (indicating an Action Unit is only observed in one context).

movements significantly associated with affiliative (non-play) contexts were blink (AU145) and half blink (AU47). Chewing (AD81) was identified in 13.5% of behaviours made in this context. Our initial ethogram included the behavioural contexts follow, contact (friendly), and groom initiation within the interaction context category of affiliative (non-play), however we did not observe enough grooming initiations ($n = 1$) for meaningful analyses. An ethogram of the facial behaviour during the remaining contexts is presented in Table 4.

### Agonistic interactions

Six different facial and head movements were frequently used in agonistic contexts overall; left and right ear flattener (EAD103L + R), left and right ear rotator (EAD104L + R), nostril dilator (AD38), and head down (AD54) (Fig. 3). The three less frequently used face and head movements significantly associated with agonistic contexts were inner brow

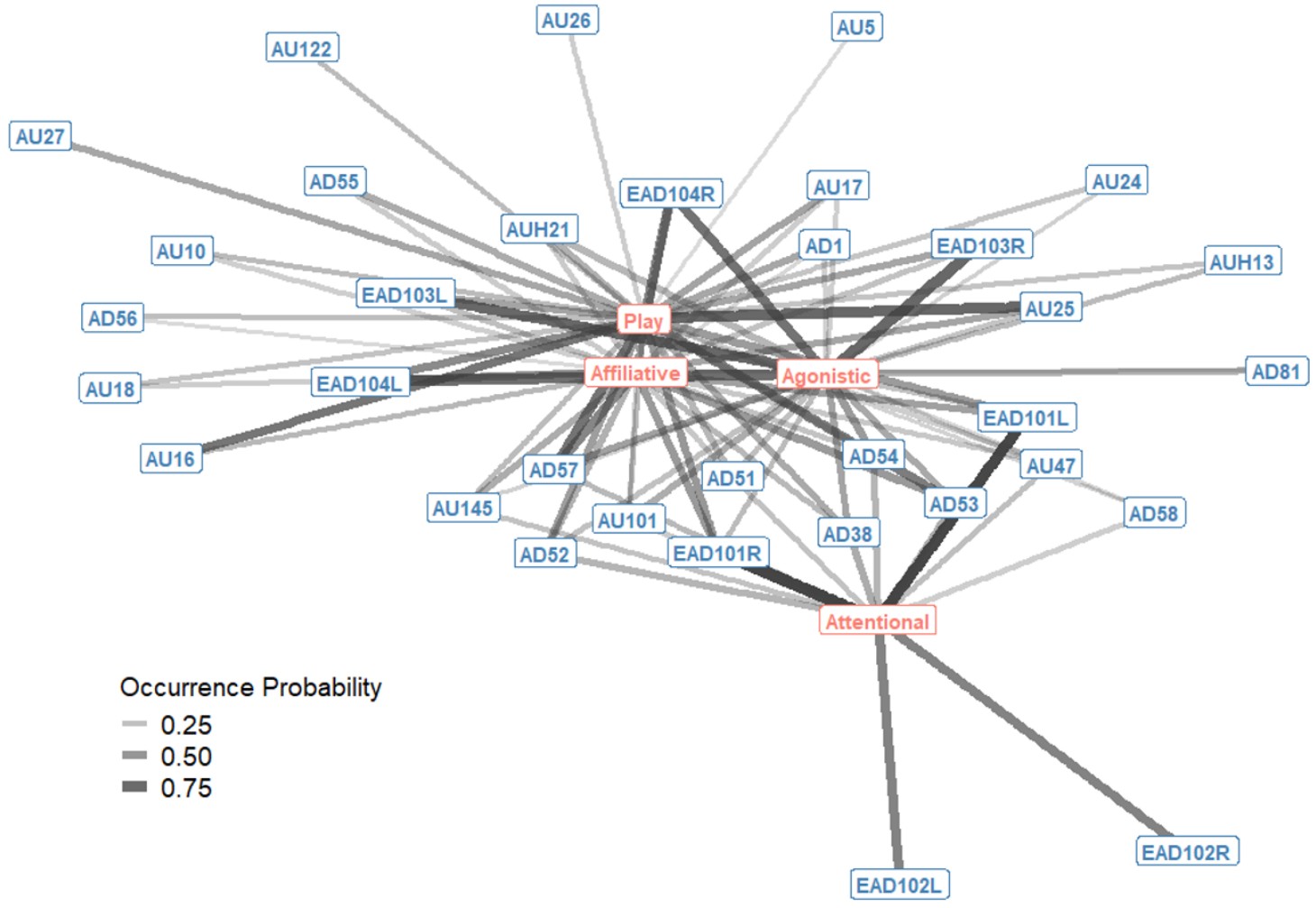

**Figure 2  Probability of occurrence bipartite network of single Action Units and Action Descriptors (blue) and interaction contexts (orange) for domestic horses.** Edges are shown for Action Units and Descriptors that occurred in at least 10% of observations per context. Edge thickness and transparency are weighted by probability of occurrence, which ranges from 0 (when an AU/AD never occurs in a context) to 1 (when an AU/AD always occurs in a context).

raiser (AU101), nostril lift (AUH13), and facial tightener (AUH21). Chewing (AD81) was identified in 33.8% of behaviours made in this context.

Our initial ethogram included the behavioural contexts displace, head threat, bite threat, kick threat, bite, kick, strike, chase, and push within the interaction context 'agonistic'. However, we did not have sufficient data to meaningfully analyse the contexts bite threat ($n = 3$), bite ($n = 2$), kick ($n = 1$), and push ($n = 1$). The facial movements associated with each of the contexts for which there were enough data are presented in Table 5.

### Attentional interactions/states

Overall, facial movements frequently used in attentional contexts were ears forward (EAD101L + R) and ears adducted (EAD102L + R) (Fig. 3). Less frequently used face and head movements significantly associated with attentional contexts were half blink (AU47),

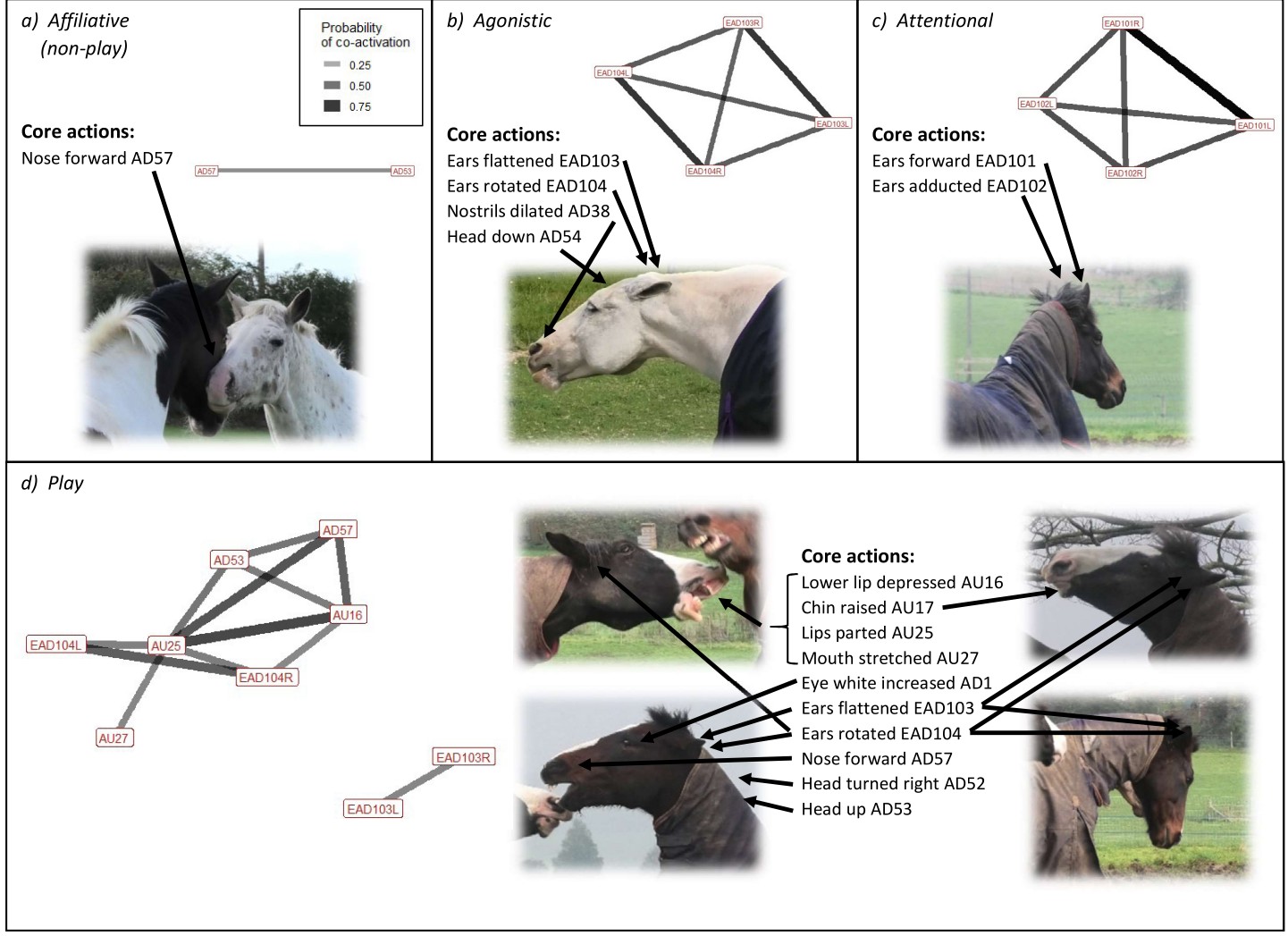

**Figure 3** **Frequently occurring Action Units (AUs) and Action Descriptors (ADs) occurring significantly more than chance ($p < 0.01$) for the four key interaction contexts of the domestic horse.** Shown alongside network plots of the frequently occurring pairs of AUs and ADs for each interaction context, and a typical example/s of facial behaviour for that context, demonstrating the use of significant AU/ADs. In network plots, edges are shown for AU/ADs occurring in ≥ 10% of observations per interaction context. Edge thickness and transparency are weighted by observed probability.

head down (AD54), and nose back (AD58). For attentional interaction contexts, the initial ethogram consisted of the behavioural contexts alert and olfactory investigation. There was sufficient data to meaningfully analyse both of these, and the facial movements associated with each can be seen in Table 6.

## Play interactions

The facial and head movements frequently used in play contexts overall were lower lip depressor (AU16), chin raiser (AU17), lips part (AU25), mouth stretch (AU27), eye white increase (AD1), ear flattener (EAD103L + R), ear rotator (EAD104L + R), head turn right (AD52), head up (AD53), and nose forward (AD57) (Fig. 3). Less frequently used face

**Table 4  Facial behaviour ethogram of horses during affiliative (non-play) horse-horse interactions.**

| Behavioural Context | Single AU/ADs significantly ($p < 0.01$) associated with context (probability of their occurrence) | Examples | AU/AD network, showing co-activation of pairs of AU/ADs. Only connections with a probability of co-activation of >0.30, and that occurred in at least 10% of observations (or at least one observation where $n < 10$) for that context are shown. | Description |
|---|---|---|---|---|
| Contact (friendly)<br>$n=132$<br>$n_{horses}=23$ | **Ears:**<br>None<br>**Eyes:**<br>AU47 (0.22)<br>AU145 (0.39)<br>**Lower face:**<br>AU10 (0.23)<br>AU18 (0.24)<br>AU25 (0.52)<br>**Head position:**<br>AD52 (0.43)<br>AD53 (0.62)<br>AD55 (0.26)<br>AD57 (0.76)<br>**Gross behaviour:**<br>None | 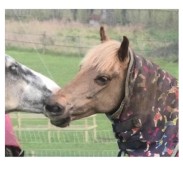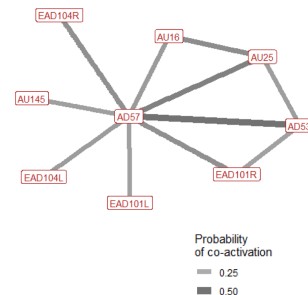 | 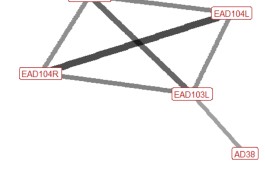<br>Probability of co-activation<br>0.25<br>0.50<br>0.75 | Facial behaviour during friendly contact is not highly specific. Instead there are a number of different movements which may be present, but the combinations in which these appear are variable. The nose is typically pushed forward. The head is often also raised, and/or may be turned right. In around half of cases of contact the lips are parted slightly. This is sometimes accompanied by raising of the upper lip to reveal a slight view of the upper teeth, or the dropping of the lower lip. There may also be some puckering of the upper lip, which extends to make contact with the other individual. Ears may be in any position and often act independently of one another. Other movements we might see are an increase in blinking and/or half blinking. |
| Follow<br>$n=45$<br>$n_{horses}=21$ | **Ears:**<br>EAD103L (0.54)<br>EAD103R (0.46)<br>EAD104L (0.60)<br>EAD104R (0.63)<br>**Eyes:**<br>None<br>**Lower face:**<br>AD38 (0.43)<br>**Head position:**<br>None<br>**Gross behaviour:**<br>AD81 (0.31) | 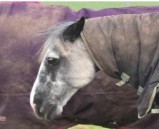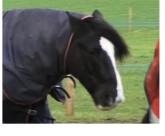 | | When following another individual, the ears are often rotated backwards and/or flattened downwards, towards the neck. The eyes remain fairly neutral, with no notable actions being utilised. The nostrils will be flared in around 40% of instances. Chewing is regularly observed during bouts of following on pasture, as this behaviour often occurs during grazing. |

and head movements significantly associated with play contexts were inner brow raiser (AU101), upper lid raiser (AU5), upper lip raiser (AU10), nostril lift (AUH13), lip pucker (AU18), upper lip curl (AU122), lip pressor (AU24), jaw drop (AU26), facial tightener (AUH21), head turn left (AD51), head tilt left (AD55), head tilt right (AD56) and nose back (AD58).

For play interaction contexts, the initial ethogram consisted of the behavioural contexts frolic, leap, run, chase (play), buck, prance, nip air, nip, bite (play), reach, grasp, neck wrestle, push (play), stamp, rear, kick threat (play), kick (play), balk (play), evasive jump, evasive spin, and head snatch. However, we did not have sufficient data to meaningfully analyse the contexts frolic ($n=0$), leap ($n=0$), chase (play) ($n=1$), buck ($n=2$), prance ($n=0$), neck wrestle ($n=2$), balk (play) ($n=0$), and evasive spin ($n=4$). The facial movements associated with each of the contexts for which there were enough data are presented in Table 7: run, nip air, nip, bite (play), reach, grasp, push (play), stamp, rear, kick threat (play), kick (play), evasive jump, head snatch.

**Table 5** Facial behaviour ethogram of horses during agonistic horse-horse interactions.

| Behavioural context | Single AU/ADs significantly (*p* < 0.01) associated with context (probability of their occurrence) | Examples | AU/AD network, showing co-activation of pairs of AU/ADs. Only connections with a probability of co-activation of >0.30, and that occurred in at least 10% of observations (or at least one observation where n<10) for that context are shown. | Description |
|---|---|---|---|---|
| Displace<br>*n*=52<br>*n_horses*=23 | **Ears:**<br>EAD103L (0.55)<br>EAD103R (0.50)<br>EAD104L (0.74)<br>EAD104R (0.70)<br>**Eyes:**<br>AU101 (0.30)<br>**Lower face:**<br>None<br>**Head position:**<br>AD54 (0.58)<br>**Gross behaviour:**<br>AD81 (0.54) | 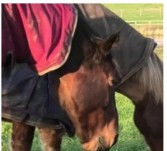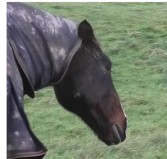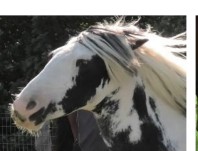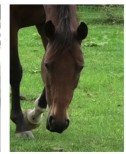 | 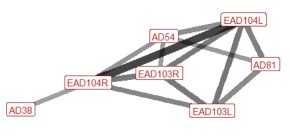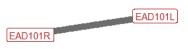 | Compared with the other agonistic signals, facial behaviour occurring during displacements uses relatively few individual actions. The ears are usually rotated backwards and/or flattened downwards, towards the neck. When the ears are flattened we may also see flaring of the nostrils. It is typical for the head to be lowered. The brow above the inner corner of the eye may be raised. Chewing is common during displace behaviour when horses are at pasture, as it often occurs during grazing. There are some instances of displace where none of these actions are seen, and instead the ears are forward. |
| Head threat<br>*n*=52<br>*n_horses*=24 | **Ears:**<br>EAD103L (1.0)<br>EAD103R (0.98)<br>EAD104L (0.62)<br>EAD104R (0.63)<br>**Eyes:**<br>AU101 (0.40)<br>**Lower face:**<br>AUH13 (0.35)<br>AD38 (0.46)<br>**Head position:**<br>AD54 (0.58)<br>AD57 (0.64)<br>**Gross behaviour:**<br>AD81 (0.37) | 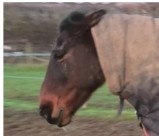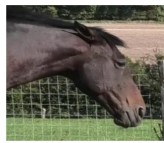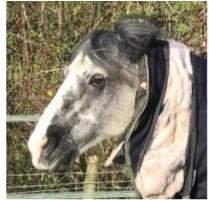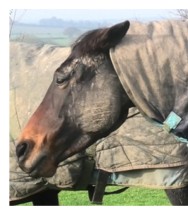 | 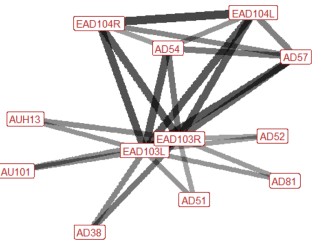 | Facial behaviour during head threats is similar to that of displace, however more AU/ADs are activated. The ears are always rotated backwards and are typically also flattened downwards, towards the neck. The head is often lowered, as in a displace, however the nose is usually also pushed forward, and the head may turn to the left or the right, towards the threat receiver. The brow above the inner corner of the eye may be raised. In the lower face, the nostrils may be flared. Chewing occurs in around a third of head threats when horses are at pasture, as it often occurs during grazing. |
| Kick threat<br>*n*=16<br>*n_horses*=13 | **Ears:**<br>EAD103L (0.83)<br>EAD103R (0.90)<br>EAD104L (0.83)<br>EAD104R (0.90)<br>**Eyes:**<br>AU101 (0.36)<br>**Lower face:**<br>AUH13 (0.36)<br>AUH21 (0.39)<br>**Head position:**<br>AD51 (0.38)<br>AD53 (0.63)<br>AD54 (0.44)<br>**Gross behaviour:**<br>None | 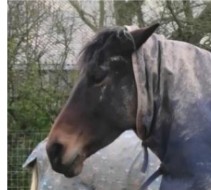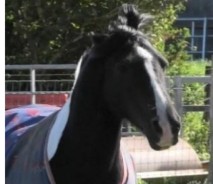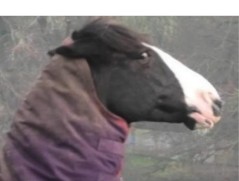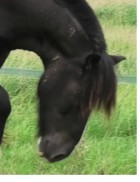 | 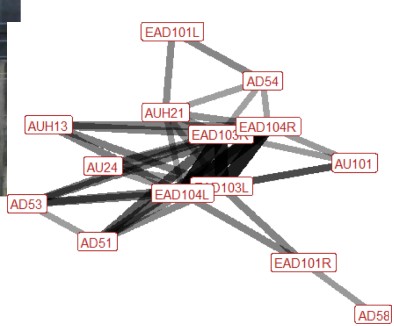 | During a kick threat the ears are almost always rotated backwards and flattened downwards, towards the neck. Sometimes one or both ears will also flick forwards during the behaviour. The brow above the inner corner of the eye may be raised, the top edges of the nostrils may be lifted in the direction of the eye, and there may be tightening across the side of the face between the eye, cheek and muzzle, making the underlying structures appear more visible. The head may be raised or lowered. |

**Table 5** (*continued*)

| Behavioural context | Single AU/ADs significantly (*p* < 0.01) associated with context (probability of their occurrence) | Examples | AU/AD network, showing co-activation of pairs of AU/ADs. Only connections with a probability of co-activation of >0.30, and that occurred in at least 10% of observations (or at least one observation where n<10) for that context are shown. | Description |
|---|---|---|---|---|
| Strike<br>*n*=15<br>*n*<sub>horses</sub>=8 | **Ears:**<br>EAD103L (0.54)<br>EAD103R (0.67)<br>EAD104L (0.85)<br>EAD104R (0.85)<br>**Eyes:**<br>AU5 (0.39)<br>AU101 (0.39)<br>AD1 (0.33)<br>**Lower face:**<br>AU18 (0.43)<br>AUH21 (0.71)<br>AD38 (0.64)<br>**Head position:**<br>AD51 (0.47)<br>AD52 (0.53)<br>AD53 (0.93)<br>AD55 (0.23)<br>AD58 (0.67)<br>**Gross behaviour:**<br>None | 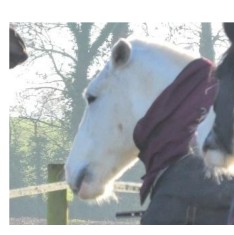 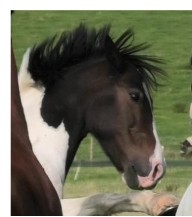 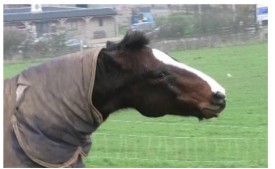 | 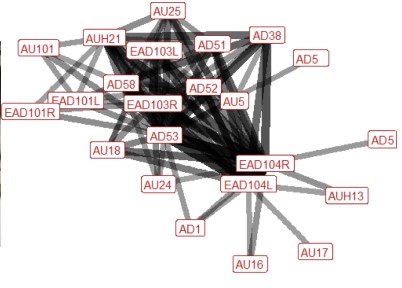 | During strikes the ears are typically rotated backwards and flattened downwards, towards the neck. The head is almost always raised, and typically turns to one side or the other during the behaviour. The nose is usually pulled back, towards the chest, but may on some occasions be pushed forwards instead. In the lower face we often see flared nostrils and tightening across the side of the face between the eye, cheek and muzzle, making the underlying structures appear more visible. In around 40% of cases we also see puckering of the upper lip. The whites of the eyes become more visible in a third of strikes. We may also see raising of the inner brow or raising of the upper eyelid across its length, widening the eye. |
| Chase<br>*n*=15<br>*n*<sub>horses</sub>=7 | **Ears:**<br>EAD103L (0.73)<br>EAD103R (0.75)<br>EAD104L (0.82)<br>EAD104R (0.67)<br>**Eyes:**<br>AU101 (0.50)<br>**Lower face:**<br>AUH13 (0.33)<br>AU17 (0.33)<br>AUH21 (0.50)<br>AD38 (0.47)<br>**Head position:**<br>AD53 (0.60)<br>**Gross behaviour:**<br>None | 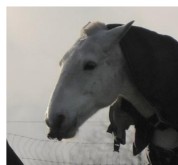 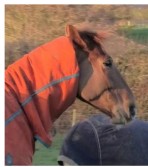 | 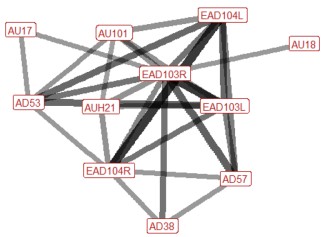 | During chasing the ears are typically rotated backwards and flattened downwards, towards the neck. The inner brow is raised in around half of occurrences. The head is typically raised, and in some instances the nose is pushed forwards. In the lower face the nostrils may be flared, or the top edges of the nostrils may be lifted in the direction of the eye. In 50% of instances there will be tightening across the side of the face between the eye, cheek and muzzle, making the underlying structures appear more visible. During chasing there is also sometimes puckering of the lip, or the area under the chin may lift upwards, making the chin look more defined. |

# DISCUSSION

This study provides the first systematic and anatomically based description of equid facial behaviours produced by horses in social contexts. In our ethogram, we describe a wide range of facial behaviours made by domestic horses during 22 behaviours related to affiliative interactions, agonistic interactions, attentional interactions/states, and during play. For these broader interaction contexts, we also identified facial movements typical of each. Nose forward frequently occurs in affiliative contexts, and in attentional contexts ears are typically forward and adducted. During agonistic interactions the ears are frequently turned backwards and flattened, the nostrils are dilated, and the head is lowered. Play involves a greater number of facial movements; the lower lip is frequently depressed, the chin raised, lips are parted and often the mouth is stretched open wide, the ears are rotated and flattened backwards, there is an increase in visible eye white, the nose is pushed forward, and the head is frequently up and/or turned to the right. Plots of context specificity and
**Table 6  Facial behaviour ethogram of horses during attentional contexts.**

| Behavioural Context | Single AU/ADs significantly ($p < 0.01$) associated with context (probability of their occurrence) | Examples | AU/AD network, showing co-activation of pairs of AU/ADs. Only connections with a probability of co-activation of >0.30, and that occurred in at least 10% of observations (or at least one observation where n<10) for that context are shown. | Description |
|---|---|---|---|---|
| Alert<br>*n=38*<br>$n_{horses}$=16 | **Ears:**<br>EAD101L (1.00)<br>EAD101R (1.00)<br>EAD102L (0.86)<br>EAD102R (0.86)<br>**Eyes:**<br>AU145 (0.32)<br>**Lower face:**<br>None<br>**Head position:**<br>AD53 (0.87)<br>**Gross behaviour:**<br>None | 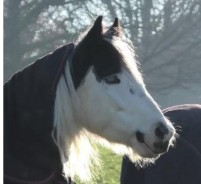 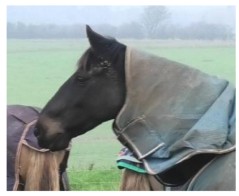 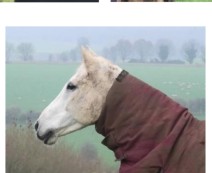 | 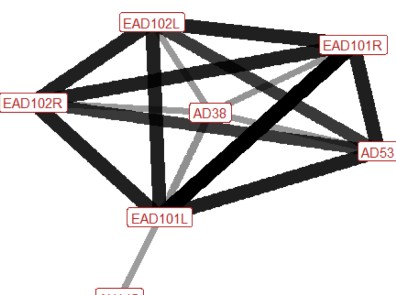 | An alert horse will have a raised head, and their ears are turned forwards and brought inwards towards each other (adducted). In around a third of alert occurrences there is also an increase in blinking. Flaring of the nostrils may sometimes be seen. |
| Olfactory investigation<br>*n=75*<br>$n_{horses}$=28 | **Ears:**<br>EAD101L (0.94)<br>EAD101R (0.94)<br>EAD102L (0.46)<br>EAD102R (0.45)<br>**Eyes:**<br>AU47 (0.34)<br>**Lower face:**<br>None<br>**Head position:**<br>AD54 (0.48)<br>AD58 (0.27)<br>**Gross behaviour:**<br>None | 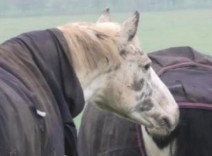 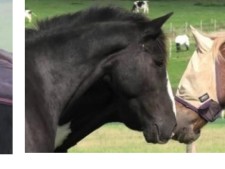 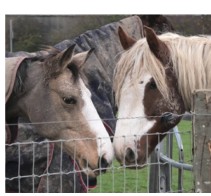 | 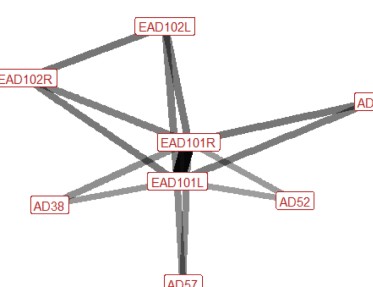 | During olfactory investigation the ears are turned forwards, and may also be brought inwards towards each other (adducted). The head is down in around 50% of cases, but can also be in other positions. In around a third of occurrences there is also an increase in half blinking. |

occurrence probability revealed that horses use the majority of AU/ADS flexibly across multiple different interaction contexts, however there are a number of AU/ADs that are highly likely to occur in each context. The within-context variation in facial behaviour observed, both within and between individuals, highlights the importance of utilising whole-body behaviours, alongside facial behaviour, when evaluating the perceived intent of social signals. We also defined a new EquiFACS Action Unit, AUH21, likely caused by activation of the platysma muscle, and which causes the muscles and structures on the side of the face to become more prominent.

The development of animal FACS in non-primate species has shown that primates are not unique in their production of complex facial behaviours. Horses have 17 defined AUs (18 with the addition of AUH21) (*Wathan et al., 2015*), greater than any other species for which FACS have been developed, with the exception of cats (21 AUs) (*Bennett, Gourkow & Mills, 2017*) and humans (30 AUs) (*Ekman & Friesen, 1978*). It is therefore not surprising that they also have an extensive facial behaviour repertoire, combining these AUs in complex arrangements dependent on context. By identifying a rich repertoire of facial movements in a species phylogenetically distinct from those previously examined,
**Table 7  Facial behaviour ethogram of horses during play contexts.**

| Behavioural Context | Single AU/ADs significantly ($p < 0.01$) associated with context (probability of their occurrence) | Examples | AU/AD network, showing co-activation of pairs of AU/ADs. Only connections with a probability of co-activation of >0.30, and that occurred in at least 10% of observations (or at least one observation where n<10) for that context are shown. | Description |
|---|---|---|---|---|
| Run<br>$n=8$<br>$n_{horses}=7$ | **Ears:**<br>EAD101L (0.63)<br>EAD104L (0.63)<br>EAD104R (0.86)<br>**Eyes:**<br>None<br>**Lower face:**<br>AD38 (0.88)<br>**Head position:**<br>AD53 (0.63)<br>**Gross behaviour:**<br>None | 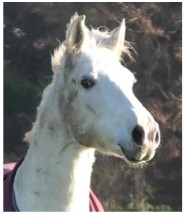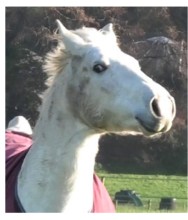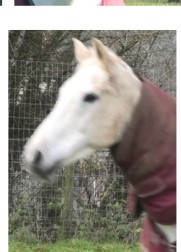 | 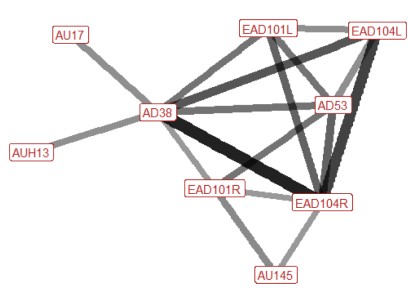 | When running the ears may be turned forwards or rotated backwards and flattened towards the head, and often they will alternate between the different ear positions. The nostrils are flared and the head is typically raised. |
| Nip air<br>$n=81$<br>$n_{horses}=17$ | **Ears:**<br>EAD103L (0.48)<br>EAD103R (0.47)<br>EAD104L (0.63)<br>EAD104R (0.64)<br>**Eyes:**<br>AD1 (0.50)<br>**Lower face:**<br>AU10 (0.30)<br>AU16 (1.00)<br>AU17 (0.68)<br>AU18 (0.28)<br>AUH21 (0.40)<br>AU24 (0.18)<br>AU25 (1.00)<br>AU26 (0.43)<br>AU27 (0.59)<br>AU122 (0.56)<br>**Head position:**<br>AD51 (0.40)<br>AD52 (0.51)<br>AD53 (0.83)<br>AD55 (0.44)<br>AD56 (0.42)<br>AD57 (0.90)<br>**Gross behaviour:**<br>None | 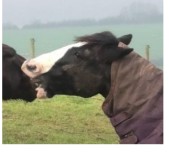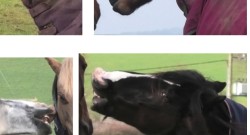 | 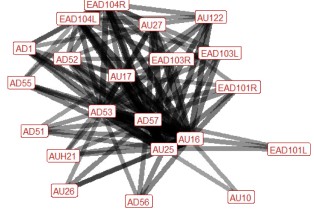 | In nip air the ear position is variable; they can be rotated backwards, flattened towards the head, or turned forwards. In around half of occurrences there will be an increase in the amount of eye white visible. In the lower face there may be numerous potential actions involved. Importantly, the lips are always parted and the lower lip is pulled down. We also typically see the area under the chin lifting upwards, making the chin look more defined, as well as the mouth being stretched wide open, and curling upwards of the upper lip. Other actions that may occur are raising of the upper lip revealing the upper teeth, puckering of the top lip, pressing together of the lips (after the nip has been completed), dropping of the lower jaw (as opposed to full stretching open of the mouth), and tightening across the side of the face between the eye, cheek and muzzle, making the underlying structures appear more visible. The head is quite mobile during this behaviour. It may be turned or twisted to the left or right, but is usually raised, with the nose pushed forwards. |

**Table 7** (*continued*)

| Behavioural Context | Single AU/ADs significantly ($p < 0.01$) associated with context (probability of their occurrence) | Examples | AU/AD network, showing co-activation of pairs of AU/ADs. Only connections with a probability of co-activation of >0.30, and that occurred in at least 10% of observations (or at least one observation where n<10) for that context are shown. | Description |
|---|---|---|---|---|
| Nip<br>*n*=59<br>*n_horses*=18 | **Ears:**<br>EAD104L (0.57)<br>**Eyes:**<br>AU101 (0.37)<br>AU145 (0.35)<br>**Lower face:**<br>AU10 (0.48)<br>AUH13 (0.26)<br>AU16 (0.98)<br>AU17 (0.30)<br>AU18 (0.20)<br>AUH21 (0.32)<br>AU24 (0.19)<br>AU25 (1.00)<br>AU26 (0.20)<br>AU27 (0.80)<br>AU122 (0.38)<br>AD38 (0.42)<br>**Head position:**<br>AD52 (0.44)<br>AD55 (0.49)<br>AD56 (0.34)<br>AD57 (0.80)<br>**Gross behaviour:**<br>None | 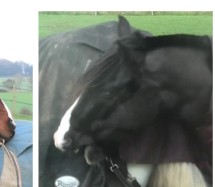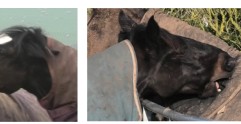 | 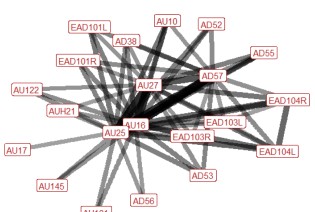 | During nip air the ear position is variable; they can be rotated backwards, flattened towards the head, or turned forwards. In a third of instances we will observe raising of the inner brow. Similarly, there is increased blinking in around a third of occurrences. As with nip air, there may be numerous potential actions involved in the lower face during nip. Importantly, the lips are always parted and the lower lip is pulled down. We also usually see the mouth stretched wide open, with jaw dropping seen instead of this more intense stretching in only a fifth of cases. Other actions that may occur are raising of the upper lip revealing the upper teeth, puckering of the top lip, pressing together of the lips (after the nip has been completed), the area under the chin lifting upwards (making the chin look more defined), curling upwards of the upper lip, and tightening across the side of the face between the eye, cheek and muzzle, making the underlying structures appear more visible. Flaring of the nostrils, or lifting of the top edges of the nostrils in the direction of the eye, may be seen. The head is usually tilted to one side (either left or right). The nose is typically pushed forwards. On occasion we may also see the head being turned right, or the head being raised. |
| Bite (play)<br>*n*=34<br>*n_horses*=9 | **Ears:**<br>None<br>**Eyes:**<br>AU47 (0.26)<br>AU101 (0.42)<br>AD1 (0.29)<br>**Lower face:**<br>AU10 (0.58)<br>AU16 (0.90)<br>AU17 (0.34)<br>AUH21 (0.38)<br>AU25 (1.00)<br>AU27 (0.91)<br>AU122 (0.39)<br>**Head position:**<br>AD55 (0.38)<br>AD56 (0.41)<br>AD57 (0.82)<br>**Gross behaviour:**<br>None | 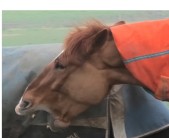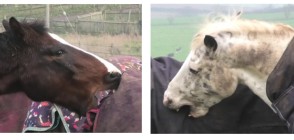 | 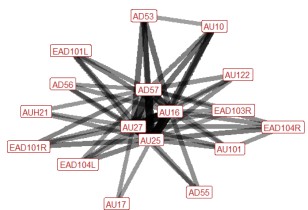 | During nip air the ear position is variable; they can be rotated backwards, flattened towards the head, or turned forwards. During play biting the mouth stretches wide open, parting the lips, and the lower lip is pulled down so that the lower teeth are visible. The upper lip is often raised to reveal the upper teeth too. Other lower facial actions that may be seen are raising of the chin to make it more pronounced, curling upwards of the upper lip, and tightening across the side of the face between the eye, cheek and muzzle, making the underlying structures appear more visible. The nose is usually pushed forwards, and is usually tilted to one side (left or right). In some instances, there may be raising of the inner brow, an increase in the amount of visible eye white, or an increase in half blinking. |
| Reach<br>*n*=26<br>*n_horses*=12 | **Ears:**<br>EAD103R (0.44)<br>**Eyes:**<br>AU47 (0.25)<br>AD1 (0.40)<br>**Lower face:**<br>AU16 (0.69)<br>AU17 (0.39)<br>AU18 (0.40)<br>AUH21 (0.46)<br>AU25 (0.78)<br>AU122 (0.36)<br>**Head position:**<br>AD51 (0.31)<br>AD52 (0.58)<br>AD53 (0.69)<br>AD55 (0.39)<br>AD56 (0.31)<br>AD57 (0.96)<br>**Gross behaviour:**<br>None | 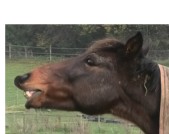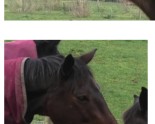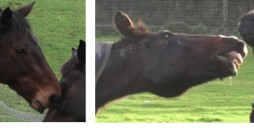 | 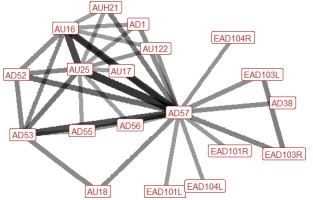 | When a horse is performing reach, the position of the ears is quite variable and the ears may move independently of each other. The nose is pushed forwards and the head is usually raised. The head is usually turned and/or twisted (both directions are possible). The lips are typically parted, with the lower lip dropped to reveal the lower teeth. This may be accompanied by raising of the chin, which creates a distinct appearance where the lip appears extended and square (see top left and bottom right plates). We may see the upper lip curling upwards or, alternatively, it may be puckered. In around half of occurrences there will be tightening across the side of the face between the eye, cheek and muzzle, making the underlying structures appear more visible. In some instances, we see an increase in visible eye white or increased half blinking. |

**Table 7** (*continued*)

| Behavioural Context | Single AU/ADs significantly ($p < 0.01$) associated with context (probability of their occurrence) | Examples | AU/AD network, showing co-activation of pairs of AU/ADs. Only connections with a probability of co-activation of >0.30, and that occurred in at least 10% of observations (or at least one observation where n<10) for that context are shown. | Description |
|---|---|---|---|---|
| Grasp<br>*n=40*<br>*n_horses=13* | **Ears:**<br>None<br>**Eyes:**<br>AU145 (0.39)<br>AD1 (0.31)<br>**Lower face:**<br>AU10 (0.24)<br>AUH13 (0.25)<br>AU24 (0.26)<br>AU25 (0.66)<br>**Head position:**<br>AD55 (0.45)<br>AD56 (0.25)<br>AD57 (0.75)<br>**Gross behaviour:**<br>None | 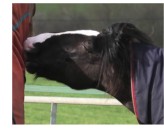 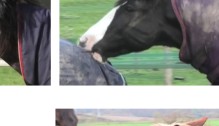 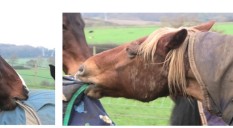 | 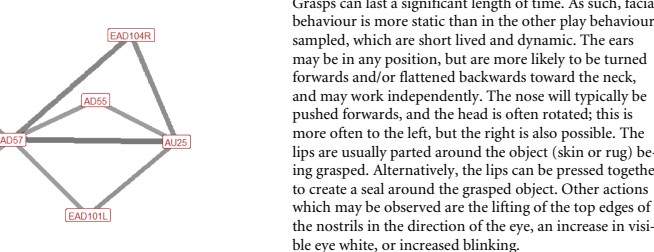 | Grasps can last a significant length of time. As such, facial behaviour is more static than in the other play behaviours sampled, which are short lived and dynamic. The ears may be in any position, but are more likely to be turned forwards and/or flattened backwards toward the neck, and may work independently. The nose will typically be pushed forwards, and the head is often rotated; this is more often to the left, but the right is also possible. The lips are usually parted around the object (skin or rug) being grasped. Alternatively, the lips can be pressed together to create a seal around the grasped object. Other actions which may be observed are the lifting of the top edges of the nostrils in the direction of the eye, an increase in visible eye white, or increased blinking. |
| Push (play)<br>*n=16*<br>*n_horses=8* | **Ears:**<br>EAD101L (0.55)<br>EAD101R (0.67)<br>EAD103L (0.60)<br>EAD103R (0.63)<br>EAD104L (0.73)<br>EAD104R (0.73)<br>**Eyes:**<br>AU145 (0.46)<br>**Lower face:**<br>AU17 (0.33)<br>AUH21 (0.44)<br>AU24 (0.40)<br>AD38 (0.44)<br>**Head position:**<br>AD51 (0.33)<br>AD52 (0.53)<br>AD54 (0.47)<br>AD58 (0.40)<br>**Gross behaviour:**<br>None | 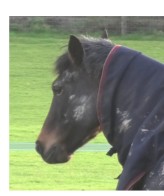 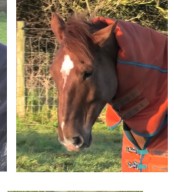 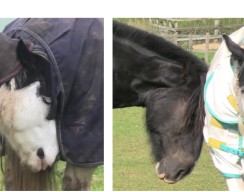 | | In push (play) the ears may be turned forward, rotated backwards, or flattened downwards towards the neck. They may move during the behaviour, so that multiple of these ear positions are seen in succession. There is an increase in blinking in around a half of instances of push (play). In the lower face, we may see the lips pressing together, raising of the chin (making it appear more pronounced), flaring of the nostrils, or tightening across the side of the face between the eye, cheek and muzzle, making the underlying structures appear more visible. The head may be up or down, and is often turned to one side. The nose is pulled back towards the chest in around 40% of instances. |
| Stamp<br>*n=10*<br>*n_horses=5* | **Ears:**<br>EAD101L (0.88)<br>EAD101R (0.80)<br>**Eyes:**<br>None<br>**Lower face:**<br>AD38 (0.50)<br>**Head position:**<br>AD53 (0.70)<br>**Gross behaviour:**<br>None | 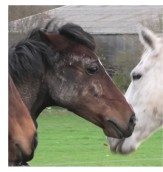 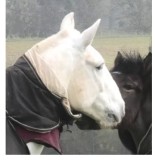 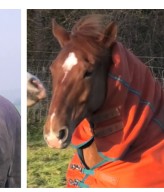 | 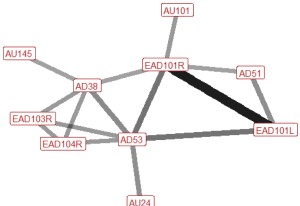 | During stamping the ears are almost always pointed forwards at some point during the behaviour, although can also be rotated and flattened backwards towards the neck. The head is typically raised, and the nostrils are flared in around a half of instances of stamp. When nostrils are flared, increased blinking may sometimes be observed. Other actions which may be seen on occasion are the raising of the inner brow and the pressing together of the lips. |

**Table 7** (*continued*)

| Behavioural Context | Single AU/ADs significantly ($p < 0.01$) associated with context (probability of their occurrence) | Examples | AU/AD network, showing co-activation of pairs of AU/ADs. Only connections with a probability of co-activation of >0.30, and that occurred in at least 10% of observations (or at least one observation where n<10) for that context are shown. | Description |
|---|---|---|---|---|
| Rear<br>*n*=12<br>*n*<sub>horses</sub>=7 | **Ears:**<br>EAD103L (0.83)<br>EAD103R (0.82)<br>EAD104L (0.75)<br>EAD104R (0.90)<br>**Eyes:**<br>AD1 (0.91)<br>**Lower face:**<br>AU16 (0.91)<br>AU17 (0.64)<br>AU18 (0.46)<br>AU25 (0.92)<br>AU27 (0.50)<br>AD38 (0.46)<br>**Head position:**<br>AD51 (0.68)<br>AD52 (0.58)<br>AD53 (0.83)<br>AD57 (0.83)<br>AD58 (0.42)<br>**Gross behaviour:**<br>None | 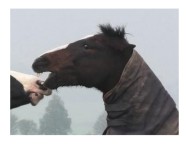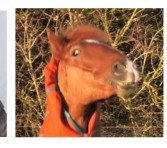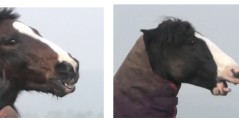 | 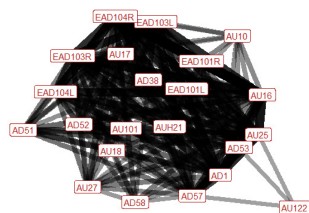 | When rearing during play the ears are typically rotated backwards and flattened downwards towards the neck at some point during the behaviour. Sometimes the ears may also rotate forwards during the behaviour. Rearing is also characterised by an increase in the amount of visible eye white, the lips being parted (with the mouth being fully stretched open in 50% of occurrences), and the lower lip being moved downwards to reveal the lower teeth. There may also be raising of the chin, making it appear more distinct. This creates a distinct appearance, where the lip appears extended and square, when it occurs in conjunction with the lowering of the lower lip (see bottom right plate). Puckering of the upper lip, and flaring of the nostrils, are also fairly common. The head is usually up, and the nose will typically be pushed forwards at some point during rearing, although it may also be brought back towards the chest within the same sequence. The head is likely to be turned to one side (left, right, or both in succession). |
| Kick threat (play)<br>*n*=16<br>*n*<sub>horses</sub>=6 | **Ears:**<br>EAD103L (0.68)<br>EAD103R (0.64)<br>EAD104L (0.81)<br>EAD104R (0.92)<br>**Eyes:**<br>AU5 (0.43)<br>AU47 (0.40)<br>AU101 (0.39)<br>AD1 (0.33)<br>**Lower face:**<br>AUH13 (0.44)<br>AU24 (0.44)<br>AD38 (0.50)<br>**Head position:**<br>AD53 (0.69)<br>AD54 (0.44)<br>AD58 (0.31)<br>**Gross behaviour:**<br>None | 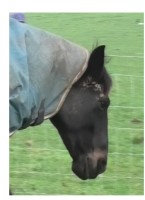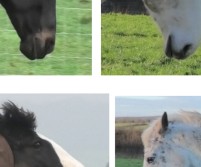 | 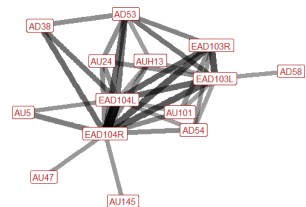 | Kick threats during play are typically defined by the ears being rotated backwards and flattened downwards towards the neck. The head is usually up, but may alternatively be down, and in some cases the head is lowered during the kick action, then raised immediately afterwards. The nose is brought in towards the chest in around a third of instances. With regards to the eyes, there may be raising of the upper eyelid (making the eye appear larger in size), increased blinking or half blinking, raising of the inner brow, or an increase in the amount of visible eye white. In the lower face we may see pressing together of the lips, flaring of the nostrils, or lifting of the top edges of the nostrils, in the direction of the eye. |
| Kick (play)<br>*n*=6<br>*n*<sub>horses</sub>=4 | **Ears:**<br>EAD104L (0.83)<br>EAD104R (1.00)<br>**Eyes:**<br>None<br>**Lower face:**<br>AD38 (1.00)<br>**Head position:**<br>None<br>**Gross behaviour:**<br>None | 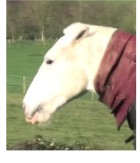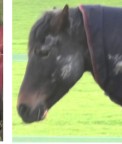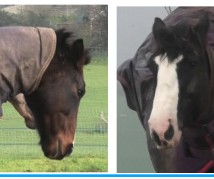 | 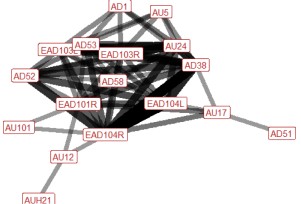 | During kick (play) the ears are typically rotated backwards, and flattened downwards towards the neck. The nostrils always flare at some point during the behaviour, and the nose is typically pulled back towards the chest. Other actions we may see include the pressing together of the lips, the head being raised, an increase in visible eye white, raising of the upper eyelid (increasing the size of the eye), raising of the chin (making the chin look more defined), or pulling back at the corners of the lips. |

**Table 7** (*continued*)

| Behavioural Context | Single AU/ADs significantly ($p < 0.01$) associated with context (probability of their occurrence) | Examples | AU/AD network, showing co-activation of pairs of AU/ADs. Only connections with a probability of co-activation of >0.30, and that occurred in at least 10% of observations (or at least one observation where n<10) for that context are shown. | Description |
|---|---|---|---|---|
| Evasive jump<br>n=24<br>$n_{horses}$=10 | **Ears:**<br>EAD101L (0.57)<br>EAD101R (0.67)<br>EAD103L (0.65)<br>EAD103R (0.53)<br>EAD104L (0.78)<br>EAD104R (0.79)<br>**Eyes:**<br>AU5 (0.37)<br>AU101 (0.39)<br>AD1 (0.91)<br>**Lower face:**<br>AU10 (0.36)<br>AUH13 (0.38)<br>AU16 (0.56)<br>AU17 (0.62)<br>AU18 (0.50)<br>AUH21 (0.47)<br>AU24 (0.36)<br>AU25 (0.79)<br>AU27 (0.35)<br>AD38 (0.50)<br>**Head position:**<br>AD51 (0.34)<br>AD52 (0.63)<br>AD53 (0.88)<br>AD55 (0.33)<br>AD57 (0.71)<br>**Gross behaviour:**<br>None | 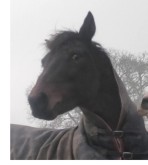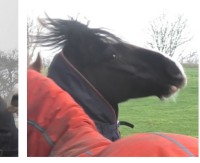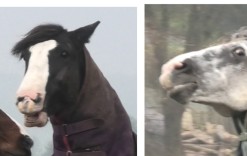 | 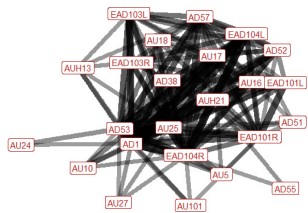 | During an evasive jump the ear position is variable; they can be rotated backwards, flattened towards the head, or turned forwards. There is almost always an increase in visible eye white, a raised head, and parted lips (in a third of instances the mouth is stretched wide open) at some point during the behaviour. The nose is typically pushed forwards and we often see the lower lip pulled downwards (to reveal a portion of the lower teeth), or raising of the chin (making the chin more defined). Sometimes these latter two occur together, creating a distinctive square, elongated shape with the lower lips (see bottom left plate). The head is typically turned to one side; more often this is to the right, but can also be to the left. Sometimes the head may tilt to the left. Regarding the eyes, we may observe raising of the inner brow, or raising of the upper eyelid, increasing the size of the eye. Lower face actions that may be seen during evasive jumps are raising and/or puckering of the upper lip, the pressing together of the lips, flaring of the nostrils, raising of the upper edge of the nostrils towards the eye, or tightening across the side of the face between the eye, cheek and muzzle, making the underlying structures appear more visible. |
| Head snatch<br>n=15<br>$n_{horses}$=8 | **Ears:**<br>EAD101L (0.87)<br>EAD101R (0.91)<br>EAD102L (0.50)<br>EAD104R (0.55)<br>**Eyes:**<br>AD1 (0.47)<br>**Lower face:**<br>AUH21 (0.40)<br>AD38 (0.43)<br>**Head position:**<br>AD51 (0.53)<br>AD52 (0.67)<br>AD53 (0.88)<br>AD55 (0.40)<br>AD58 (0.47)<br>**Gross behaviour:**<br>None | 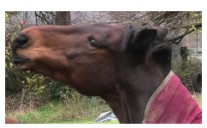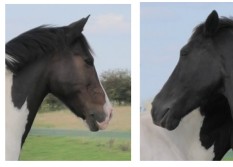 | 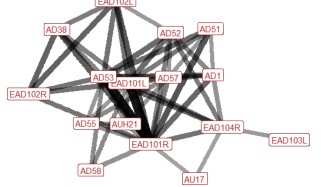 | The ears are almost always turned forwards at some point during head snatching. This may be accompanied by the bringing together of the ears centrally. Sometimes one or both ears may be rotated backwards and/or flattened down and back, towards the neck. The head is usually raised, and is often turned to one side (left or right, with right being more common). In around half of occurrences the nose is pulled backwards towards the chest, and there is sometimes tilting of the head to the left. There is an increase in visible eye white in roughly half of instances (particularly when ears are flattened), and we may see flaring of the nostrils or tightening across the side of the face between the eye, cheek and muzzle, making the underlying structures appear more visible. |

our results contribute to a growing body of evidence suggesting that socio-ecological factors have had a significant influence on the evolution of facial behaviour across multiple taxa (*Waller, Julle-Daniere & Micheletta, 2020*; *Waller & Micheletta, 2013*). While this is a comprehensive ethogram of captive horses, the full range of horse facial behaviours is likely to be wider when we consider sexual interactions, male herding behaviours, adult-young interactions and rarer/lesser performed behavioural interactions.

During affiliative interactions facial behaviour was not highly stereotyped and was relatively low in specificity. As such, there were no defining features that indicated affiliation overall (Fig. 3). This is echoed in the academic literature, where it has not been possible to identify specific facial movements associated with positive anticipation (*Ricci-Bonot & Mills, 2023*), and where low-level affiliative behaviours in horses, such as friendly contact, are rarely described, perhaps because they lack defining features. Indeed, on many

occasions it was observed that the face remained fairly neutral during these interactions. This differs from primates, where particular facial movements function to indicate that the proceeding interaction will be affiliative (*e.g.*, *Clark et al., 2022*; *Micheletta et al., 2013*; *Waller & Dunbar, 2005*). Perhaps the horse has little need for a clear benign intent signal, and it is the absence of any other, more stereotyped, signals that indicates an interaction is affiliative. Indeed, horses do not differentiate between positive anticipation facial behaviour and neutral facial behaviour (*Wathan et al., 2016*), suggesting they interpret a lack of facial movement as of positive valence. Alternatively, the communication of affiliative intent could be *via* non-visual means such as olfaction. In the equine industry, the ears being forward is seen as a marker of positive emotional state, whereas ears back and/or flattened indicates aggression, discomfort or pain. Here, however, we found that during conspecific affiliative interactions there is relatively little use of ears forward, and that the ears are often back and/or flattened (Table 4). We therefore recommend that care is taken to not automatically attribute ears back or flattened to aggressive intent or pain. It is important to consider the wider context of the behaviour, movements in other regions of the face, and whole-body posture and behaviours, in order to establish the emotional context of a horse.

Agonistic facial behaviour was characterised by flattened, backwards-facing ears and the inner brow being raised, dilated and/or lifted nostrils were also common (Fig. 3, Table 5), supporting descriptions of agonistic behaviours reported in previous equid ethograms (*Waring, 2003*). In other species with mobile ears, backward ear position is often associated with negative or fearful states (*e.g.*, *Beerda et al., 1997*; *Boissy et al., 2011*; *Fox, 1970*), and in some species with aggression (*Defensor et al., 2012*; *Pruitt, 1976*). Interestingly, ear AU/ADs do not feature in the majority of primate FACS, and humans and apes do not move their ears to communicate, although they do possess some vestigial neural circuits that suggest pinna movement was possible ancestrally (*Hackley, 2015*). Instead movements in the lower face are critical signals of aggression in primates (*Waller & Micheletta, 2013*). This is likely an adaptation that arose as the ears became less animate in this order, whereas in species, such as horses, which retained mobile pinna, ear movement remains an important signal of intent (*Wathan et al., 2015*).

Attentional facial behaviours direct the sensory organs towards a given stimulus. As we might expect, previous equid ethograms describe vigilance behaviour as involving highly specific and stereotyped head, ear and face movements (*e.g.*, *Mills & Davenport, 2002*; *Mills & Riezebos, 2005*). Here, we found that ears forward and adducted were significantly associated with attentional contexts, in line with previous descriptions of ear position (Fig. 3). Although some head positions were significantly associated with attentional behaviour, these occurred less frequently and differed between the two behaviours alert and olfactory investigation (Table 6). Mirroring existing equine ethograms, the head was usually up when horses were alert and down during olfactory investigation, allowing the horse to make use of vision and olfaction respectively. While these facial behaviours likely did not evolve as communicative signals, evidence suggests that head orientation, as well as eye and ear positions, act as visual cues for conspecifics, allowing them to locate a stimulus by observing the attentional signals of others (*Wathan & McComb, 2014*).

Blinking and half blinking were also more likely to occur during attentional states/interactions (Table 6). Vigilance and investigative behaviours allow the individual to identify potential dangers/threats, and prepare the autonomic nervous system for fight or flight (*Rørvang, Nielsen & McLean, 2020*). Being vigilant, increases in spontaneous blink rate (SBR) and in eyelid twitching (akin to a half-blink) are used as indicators of stress and/or fear in horses (*Lelláková et al., 2023*; *Merkies et al., 2019*; *Mott, Hawthorne & McBride, 2020*; *Rashid et al., 2020*; *Rørvang & Christensen, 2018*; *Young et al., 2012*), although it should be noted that in one study decreased blink rate was associated with stress (*Merkies et al., 2019*), it may be that the apparent changes in blink rate seen here reflect arousal, however, recent experimental work suggests that attention actually reduces blink rates in horses, and this same phenomenon has been observed across a range of prey species (*Cherry et al., 2020*; *Tomberg, Petagna & de Selliers de Moranville, 2024*). The different outcome observed in the present study may be due to differences in the targets of attention, or the social context. Since our data is derived from 1s intervals and was not designed to explore blink rate explicitly, a follow up study is required to better identify changes in blink rates during a range of different attentional behaviours (*e.g.*, towards a conspecific, food, or a predator) in horses, and to understand the underlying mechanisms for these changes.

Play faces were varied and dynamic (Table 7). Several play faces, particularly nip air and reach, typically involve an open mouth, lower lip depression and a raised chin. Unlike horse agonistic facial behaviours, these open-mouthed, exposed teeth, play facial configurations show marked similarities to play configurations that are ubiquitous in primate play faces (*e.g.*, *Davila Ross, Menzler & Zimmermann, 2008*; *Palagi & Mancini, 2011*; *Pellis & Pellis, 1997*; *Scopa & Palagi, 2016*; *Waller & Cherry, 2012*), and common in carnivores (*Bekoff, 1974*; *Llamazares-Martín et al., 2017*; *Nolfo, Casetta & Palagi, 2022*). Based on the AUs present it has been suggested, and supported by our findings, that these open-mouthed faces are homologous across primates, carnivores and equids, adding support to the argument that the open mouth play face is deeply rooted in mammalian biology (*Davila-Ross & Palagi, 2022*). We suggest that 'open mouth play face' behaviour is defined and utilised in equine play ethograms going forward, in order to promote and facilitate future cross-species comparisons.

There is one pair of behavioural contexts for which it is possible, from our dataset, to compare the form of agnostic and play versions of the same behaviour; kick threat (Table 5) and kick threat (play) (Table 7). These direct comparisons are of particular interest as they help us to understand more about the evolution of complex communication by comparing the signals produced in homologous social interactions which vary in their riskiness. In this case, in both behavioural contexts the ears behaved in the same manner, however there were a greater number of lower face and eye AU/ADs utilised during play. Utilising a large number of facial movements during play may make the signal more conspicuous, and previous work in primates has similarly identified play facial expressions to use a large number of facial movements (*Davila-Ross & Palagi, 2022*). A primary function of play facial expressions might be to avoid escalation and aggression by signalling that the interaction is playful. This aids in coordinating actions between playmates and, importantly, also likely prevents

escalation into real fights where there is a risk of injury (*Davila-Ross & Palagi, 2022*). That we see more conspicuous signals during play fighting 'kick threats' than in agnostic 'kick threats' suggests that, in domestic horses, play may be riskier than agonism. Conspicuous signals are less likely to be misinterpreted, and preventing misinterpretation is more crucial in riskier situations. That play is riskier than agonism is perhaps counterintuitive, however may be a consequence of the way domestic horses are managed. Agonistic expressions are a signal of aggressive intent and, as such, the accompanying risk of physical aggression is high (*Goos & Silverman, 2002*). It is therefore essential that a threat facial expression is unambiguous, allowing the receiver the opportunity to alter their behaviour and avoid a costly fight. The horses in this study were kept in same-sex groups, the mares were non-breeding, and all males were castrated. Mares tend to be most aggressive following parturition, when they have a foal at foot (*Van Dierendonck, De Vries & Schilder, 1995*; *Wells & von Goldschmidt-Rothschild, 1979*), and castrated males are less aggressive than breeding males (*Francis et al., 1992*; *Hume & Wynne-Edwards, 2005*). Rates of injurious aggression were therefore likely relatively low in the study population, allowing for lower conspicuousness during agonistic interactions. Moving forward, it would be useful to make further comparisons between homologous behaviours across contexts in order to further understand how and why these signals differ.

Whilst conducting this study, we defined a new AU for EquiFACS; AUH21, facial tightener (Table 3). AU21 is recorded in humans (*Ekman & Friesen, 1978*) and gibbons (*Waller & Cherry, 2012*), however has not been identified in any of the other species for which FACS has been developed. AUH21 appears to have the same muscular basis as AU21, the platysma, a large superficial muscle of the neck and lower face that is particularly large in humans and horses (*Naldaiz-Gastesi et al., 2018*). Interestingly, other species for which FACS have been developed also have platysma muscles, and the platysma is involved in other AU/ADS in dogs (*Waller et al., 2013*), however an associated independent AU has not been documented in these animals. The visual appearance of AU21 in horses, is different to humans due to the anatomical differences between the species. In humans and gibbons, AU21 is known as neck tightener, as it is the neck where this AU is observed in these species. In horses this tightening instead occurs across the face, hence the adaptation of the name. The face tension seen during activation of AUH21 has been recognised in horses previously, however the muscular basis has never been considered. The Horse Grimace Scale, which is used extensively in equine veterinary medicine and research for diagnosing pain, describes 'prominent strained chewing muscles' (*Dalla Costa et al., 2014*). This appears to be analogous to the presence of AUH21. Given its importance in pain, as well as its significance in both agonistic and play facial behaviours seen in the present study, the addition of AUH21 to EquiFACS is of critical importance if we are to fully appreciate the complete facial repertoire of the horse. We thus propose that future studies using EquiFACS consider including AUH21 in their analyses, despite it not quite meeting the reliability score recommended for existing FACS AUs and ADs in this instance. Coders should be aware that visibility of AUH21 can be markedly affected by lighting conditions and the horse's coat, in a way that other AUs are not. These issues will be minimised, however, when studies are conducted in controlled conditions, and we would expect to

see subsequent improvements in reliability. Such controlled studies could also refine the description of AUH21 to further improve reliability. Another consideration for future EquiFACS work is that in less controlled environments, such as in the current study, AD133 (blow) and AD38 (nostril dilator) are often indistinguishable in video footage. In agreement, recent work using EquiFACS in plains zebra (*Equus quagga*) did not code either AD, due to lack of confidence in accurately assessing them from video footage (*Hex & Rubenstein, 2024*). The exhale associated with blow is not audible when cameras are situated outdoors, particularly when conditions are wet or windy. We recommend combining these two AUs in similar situations, as we did here, to enhance reliability and prevent misattribution.

A number of, non-mutually exclusive socio-ecological factors may have led to the rich facial repertoire of horses. Firstly, horses live in complex, multi-level, societies where possession of a wide range of facial behaviours may be advantageous in managing the complex relationships and group cohesion (*Freeberg, Dunbar & Ord, 2012*; *Maeda et al., 2021*; *Waller, Julle-Daniere & Micheletta, 2020*; *Waring, 2003*). Facial mobility in primates is positively associated with group size (*Dobson, 2009*) and within-group social tolerance (*Dobson, 2012*; *Rincon et al., 2023*), suggesting the evolution of the rich facial repertoire of horses may also be socially driven. Secondly, wild, feral and free-ranging horses typically inhabit open areas such as steppes and grasslands where visibility is high and a reliance on visual signals, particularly for a prey animal, would be adaptive (*Linklater, Cameron & Stafford, 2000*; *Ransom, 2012*; *Waring, 2003*). There are a number of extant wild equid species which share similar facial morphology and somewhat similar ecologies with the horse, while varying significantly in their social systems. Indeed, EquiFACs has already been used to document multimodal communication in plains zebra (*Hex & Rubenstein, 2024*). It is therefore quite possible, with some minor modifications to EquiFACS, to test the hypotheses that social complexity and/or habitat openness have driven the evolution of facial repertoire in this clade. Finally, human selection has influenced facial musculature and movements in domestic dogs and it may be that selection for horses with more distinct facial movements (and therefore more predictable behaviours) could also have shaped the horse facial repertoire during domestication (*Kaminski et al., 2019*; *Waller et al., 2013*). Comparative studies of the facial morphology and movements of a range of wild and domesticated equid species would help to elucidate the evolutionary origins of facial behaviours across equids.

## CONCLUSIONS

Based on the systematic measurement of individual muscle movements, we have provided the first comprehensive ethogram showing that the domestic horse is capable of producing a wide range of distinct facial behaviours. We found marked similarities between the play faces of horses and the open mouth play faces observed in primates and carnivores, adding weight to the hypothesis that these facial behaviours are deep rooted in mammalian biology. In contrast, agonistic facial behaviour in the horse differs substantially from those of primates, likely due to the lack of pinna mobility in the latter leading to the development

of agonistic faces involving movement of the lower face rather than the ears. If we are to gain a comprehensive understanding of the form and function of facial behaviour more widely, it is imperative that we look beyond our closely related primate cousins to more phylogenetically distant species. Identifying the form of facial behaviour in horses will also be invaluable in future work exploring equine welfare, social behaviour, and perception, with direct applications for those working with horses.

## ACKNOWLEDGEMENTS

We thank all the staff and horses at Sparsholt College Equine Centre, Winchester, UK, for their support with data collection. Thanks also go to Catherine Morrish for her help with data collection, and to Christina Panagiotou and Sasha Donnier for their second coding of video footage. Finally, thank you to Anne Burrows for her invaluable guidance and feedback regarding AUH21 and its muscular basis.

### Funding
This study was funded by a Doctoral studentship awarded to Kate Lewis by the University of Portsmouth. The funders had no role in study design, data collection and analysis, decision to publish, or preparation of the manuscript.

### Grant Disclosures
The following grant information was disclosed by the authors:
Doctoral studentship awarded to Kate Lewis by the University of Portsmouth.

### Competing Interests
Matthew O. Parker is an Academic Editor for PeerJ.

### Author Contributions
- Kate Lewis conceived and designed the experiments, performed the experiments, analyzed the data, prepared figures and/or tables, authored or reviewed drafts of the article, and approved the final draft.
- Sebastian D. McBride conceived and designed the experiments, authored or reviewed drafts of the article, and approved the final draft.
- Jérôme Micheletta conceived and designed the experiments, authored or reviewed drafts of the article, and approved the final draft.
- Matthew O. Parker conceived and designed the experiments, authored or reviewed drafts of the article, and approved the final draft.
- Alan V. Rincon conceived and designed the experiments, analyzed the data, authored or reviewed drafts of the article, management of, and changes to, R package, and approved the final draft.
- Jen Wathan conceived and designed the experiments, authored or reviewed drafts of the article, and approved the final draft.

- Leanne Proops conceived and designed the experiments, authored or reviewed drafts of the article, and approved the final draft.

## Animal Ethics

The following information was supplied relating to ethical approvals (i.e., approving body and any reference numbers):

The study was approved by the University of Portsmouth Animal Welfare and Ethical Review Body (Application no. 1219C).

## Data Availability

The data and code are available at OSF: Lewis, Kate. 2025. "An Ethogram of Facial Behaviour in Domestic Horses (Equus Caballus), and Testing the Social Risk Hypothesis Using This Facial Behaviour Data." OSF. January 23. doi: 10.17605/OSF.IO/ZMSVX.

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
