# Peer review of "An ethogram of facial behaviour in domestic horses: evolutionary perspectives on form and function"

_PeerJ, doi:10.7717/peerj.19309_

## Round 0.1 · original submission · Minor Revisions

Dear authors, we have received reviews for your manuscript and they are very positive. Although extensive in some cases, Minor suggestions have been provided by each reviewer which I am sure you will not have any problem to address. I am looking forward to receive a new version of this manuscript.

Reviewer 1 ·

Basic reporting

Well-written manuscript that has been carefully edited - thank you to the authors for this

This is important material (reading facial 'language' in horses) and a fair number of studies are coming together in slightly different ways, slightly different timelines. I am excited to see where the equine behaviour community will find itself in the next year or two.

The Figures/Graphs/Tables/Photos are well done and described in a high level of detail

The references are quite thorough and show good depth of understanding of this and related topic areas

Where alternative explanations exist, they have been mentioned

<normally, I attach a file with edits contained within; I truly did not discover any of these normal small edits>

Experimental design

The design is well described.
Research questions are clearly presented.
I found no concerns with ethical acceptability.
**I will mention that the copy of the ethics letter that I received appeared to be from 2019 and pertain to a different study (?) Please double check

The methodology of analysis is complicated, and I'm not sure I could conduct it flawlessly without a fair bit of help, but sufficient detail is presented to pursue this particular methodology is myself or another reader chose to pursue.

Validity of the findings

I could not consider statistical analysis to be my top-level area of expertise, but this section reads well. There are no 'red flags' to indicate that findings are not valid.

Additional comments

To reiterate, the authors have put together a nicely written manuscript that has been carefully edited. This information is a good, important addition to materials currently in the equine behaviour literature. The references have been carefully chosen and the data presented will be important to furthering this topic area.

Reviewer 2 ·

Basic reporting

The manuscript is clear, with professional english throughout. The literature references provide sufficient context, and the article is well-structured with relevant figures and tables. I particularly appreciate the inclusion of raw data and R packages in the text, which enhance the study’s transparency and reproducibility. The results are well-aligned with the hypotheses, making the manuscript a valuable contribution to the field.

Experimental design

Lewis et al. present, for the first time, a comprehensive ethogram of facial expressions in domestic horses. Although the study is largely descriptive (with NetFACS being the only analysis used), I find the manuscript well-executed and easy to follow on the first read. While the results are not groundbreaking, they are nonetheless very interesting, particularly when compared to other mammalian species (primates and canids). Therefore, I have provided below a series of specific notes that I believe could improve the current state of your manuscript.
• Title: I would suggest replacing the semicolon (;) with a colon (:).
• L24: Please italicize the scientific name of the species.
• L37: Typically, Action Units (AUs) have full names. If you have assigned a full name to this new AU, I recommend including it here in parentheses.
• L62-64: Pay attention to the parentheses; they are not correctly formatted.
• L80: Either here or in a more appropriate section, I suggest emphasizing that this statement holds particularly true for domestic animals used in sports, such as horses.
• L119: I would add a brief explanation of why you chose the horse specifically and not another species, such as the donkey.
• L129: It would be helpful to provide more information on how the animals are normally kept outside of your study. Are they used to being in groups in grazing areas? Are they typically housed in stalls?
• L160: "Approximate time" is too vague; please be more precise.

In the Materials and Methods section, I would like to see more details regarding the context. How did you distinguish a playful context from an aggressive one? What did you do when the context was ambiguous? How much time after the end of a context was a facial expression still considered part of that context? Did the context of the facial expression refer to the moment during, just before, or just after the event? These details are crucial for ensuring the repeatability of the study.
It is also important to add definitions of what is meant by "playful/affiliative context" in this section, rather than leaving it for the results or discussion.

Additionally, I would appreciate more information on how you coded AUH21 and, particularly, which manuals you consulted to determine which facial muscles are involved in this movement.

• L260: I understand that the reliability score was slightly below the threshold, but I believe this aspect could be further developed. You could have repeated the reliability test several times during the video analysis, at the beginning and end of the process, to show a trend of improvement. In any case, it may be worth re-running the reliability test with a larger sample at this point.

Validity of the findings

The discussion is generally well-written, but I find it somewhat wordy and meandering, which detracts from the overall focus of the work. I would recommend making it more concise and to the point. Additionally, I suggest referencing the figures more directly within the discussion.

·

Basic reporting

Congratulations on excellent work, this is an area that needed someone to put in the hours (and I am sure there will have been many) to document facial behaviour during social interactions. I have a few areas that could be addressed to improve the ms further:

• Line 23 and 54 – we would normally talk about the animals cognitive, emotional and physical state (or health) and how the combination of the three results in behaviour – i.e. motor output. I don’t think the reference really back up the ability to say facial behaviour gives insight into the animals cognitive, emotional and behavioural state, more so as when referring to cognition in people they describe higher cognitive processes such as deception that horses are not capable of. It also becomes confusing to say facial behaviour gives insight into behavioural state. Fundamentally we are using facial behaviour to gain insight into how an animal might be feeling (and to predict how they might respond next). I think this could be reworded for clarity. Alternatively, you would need to specify what you mean by cognitive, emotional and behavioural states separately.
• Line 32/33 - would establish the FACIAL movements be more accurate and add clarity, so that people don’t get confused with gross motor movements such as rearing?
• Introduction – would there be any merit in explaining the difference between facial expression and facial behaviour? It may help set the context for the reader?
• Table 1 – You can’t change it now as this was the description of the behaviours your analysis was based on. But I found it interesting that a kick is described as ‘apparent intention to make contact’. I would suggest that intention to make contact is very rare (and if they do intend to then they usually do make contact). Even when they do make contact there is often a degree of kick inhibition – initially a low amount of force is used. I would observe kick threats make up at least 90% of these interactions.
• Line 164-171 – I can’t quite follow what you did here, no doubt my ignorance but I suspect as you understand the process you undertook it is simple for you, but harder for someone to understand that did not undertake the analysis. Can you please reword this? Also, whilst I am happy to acknowledge I have limited knowledge of network analysis, it would be expected that certain horses would contribute to the model (especially for specific behaviours) more than others – can this not be accounted for in the analysis as we would do if building a statistical model? Second also, I was particularly confused about suggesting that a maximum of 5 facial behaviours were selected for certain individuals/behaviours. I feel it would be important to code all of the facial behaviours that occur?
• Line 253 – Excellent observation and proposal for AUH21. It justifies further work to see if it can be coded with better agreement, but even if it can’t the cross species relevance makes this an interesting point. This was also really nicely discussed in the discussion section.
• Line 299 – You only observed 1 grooming bout during all of these interactions? That is worthy of further discussion as it is normally frequently seen. Might it be because many of the horses had rugs on and so it was difficult for them to participate in mutual grooming? If so this should be acknowledged as a potential welfare concern regarding rugging (obviously there are many more benefits). But mutually grooming is important to horses and I think this warrants further discussion
• Table 4 – In the 4th column you state that only those with a co-occurrence of >0.3 are included. But then include those with a co-activation on 0.25. If co-activation and co-occurrence are different can this please be explained? If not I don’t understand how you have included those with 0.25 probability.
• Table 4 – Can you explain in the discussion why AU/ADs can occur significantly together (column 4) but are not significantly associated with that context so do not appear in column 2
• Table 5 – Many of the network diagrams in column have become too complex to evaluate
• Line 463 – I don’t follow your reasoning why play may be riskier than agonism? Can you provide clarity
• You have acknowledged that facial behaviour can vary considerably for the same behavioural interaction, within and between horses, is it worth commenting that evaluation of the perceived intent of the behaviour should be based on a whole horse evaluation, rather than just facial behaviour?
• Is it worth having a discussion section on some of the limitations of the study?

Experimental design

no comment

Validity of the findings

no comment

Additional comments

minor suggestions made regarding basic reporting. Some of these also cross over with experimental design and/or validity so apologies if they are not in the correct boxes. But it felt easier to raise them chronologically.

·

Basic reporting

Abstract:
L24: Write “Equus caballus” in italic
L38: The AUH21 is described as a “movement”; yet in Results L263, it is described as “it is not based on a specific movement of part of the face ”. Is it a movement or not ? I would suggest to make this point more clear.

Introduction: is well structured.
Some items I would suggest:
1) L46-47 : In humans, this topic is mainly related to «facial expressions». Yet in others animals, you and some others colleagues addressed their concern about this designation and adequately suggested to use a more neutral word «facial behaviour» (your MS Waller et al. 2020). I would suggest that you briefly mention this shift as it could help to support your inter-species discussion and help a reader, e.g. used to human facial expressions research, to in depth understanding the topic addressed in your MS.
2) L 62: “Canis lupus familiaris » instead of « canis familiaris »)

Methods:
L131-133: Could you make it more clear what your experimental setup is ?
L137: “The group … was pseudo-randomly selected” : how was it performed ? Did you use an application ?
L146: Could you explain how you managed to have a high quality of image ? How far was the camcorder from the horses ? Did you use a zoom ? This would be interesting to provide suggestions for FB field recordings.
L161-171: These information are related to the facial behaviour analysis, not the behavioural context analysis. I suggest to move and integrate this under the § describing how facial behaviours were analysed.
L210: “… which occurred significantly (P < 0.01) more often than chance”. Could you explain this a little more? How is set the chance threshold ?

Results:
L 253: This is really interesting!
L 258: The pictures of table 3 do not help to understand the prominence of the structures.
I would suggest to add 1) a fig. illustrating the muscle insertions to help identify the location of the muscle (as there are some other muscles in the region highlighted in red in table 3) and 2) a video of the AUH21/ platysma contraction if possible or at least a combined figure including AUH21 sided by a neutral picture without AUH21, maybe with arrows to help identify what to be looked at.
L 258: In Table 3, an additional item I would suggest to add under consideration is a possible problem with the tooths (which may frequently happen and so need to be checked before video analysis of facial movements): if the cheek is injured, e.g. by a sharp tooth, the horse may manage to insert pieces of hay/straw between the tooth and the jaw and this appear externally as a more prominent cheek (could also potentially be located in your red region). I think this could help further coding of AUH21.
I would also suggest that you add some exclusion for coding AUH21 as there are some other muscles located on the side of the face in the region highlighted in red in your table.
L260: “using the same formula as previously” : which one (not yet mention in the Results)? Maybe “as described in the methods” ?
L263: See comment for the abstract: is it a movement or not ? Please make this more clear.
L263: For a more comprehensive reading, I would suggest that you briefly inform what is the function of this muscle in horses (even if you mention this in the discussion, it would help to have it already here).
L270: “between 36 domestic horses, in two established social groups”: this should be in the Methods
L274-277: This have yet be described in the Methods. If, for didactical reasons, you want to remind what context specificity is, I would suggest to more summarize lines 274 to 277 in order not to lose your reader (eager to learn about your results : )). Idem lines 286-289.
L282: “the ears being forward (EAD102L+R): Could you check this/make it clearer as in EquiFACS ears forward are coded with EAD101 (as also indicated in your table2). EAD102 (ear adductor) is coded for rotation of ear toward the midline. Yet attention may be oriented forward (EAD101), laterally (EAD102) or backward (EAD104).
L296: “two less frequently used“: this is quite confusing: could be the two less expressed among all or (I guess) this means less than 40 % of facial behaviours occurrence yet significantly associated with the behavioural context ? Maybe you could remind your threshold of 40 % in ( ) to make the reading easier.
L302: In Table 4: could you explain why AU16, which is co-activated in the contact context above 0.3 probability of occurrence, is not listed with the AUs/ADs associated with this context ? Does it mean that when AU16 is expressed, it is more than 30 % together with AU25 or AD57 yet AU16 doesn’t occurred above chance in this context behavior? I think it could be interesting to more explain as other AU/ADs have the same profile in other behavioural contexts.
L320: I'm intrigued by the relatively strong probability of co-activation of EAD101R and EAD102R (same for EAD101L and EAD102L) as the ear could not be in both positions at the same time (forward oriented and pulled towards the midline at the same time). Or were they linked by a sequential action ? Could you more explain this ?

Discussion:
L 346: Well conducted discussion
L380 : Did you measure the intensity of the facial behaviour ? This the first time that «intensity of facial behaviour» is mentioned (not mentioned in methods nor in results) yet the results were about occurrences. May be you mean few FBs expressed in term of diversity and occurrences ? I would suggest to improve this ?
L395-396: “…care is taken to not automatically attribute ears back or flattened to aggressive intent or pain”: You are completely right !
L 426: Could you more discuss your result ?
I would suggest some matter of discussion :
-you analyzed shots of 1 sec, which is very short relative to the range of inter blink intervals. Also do not forget that the time distribution of SEBs are not random but there are some time patterns of occurrence. For example, SEB tend to happen when the vision is disturbed like in head displacements, gaze shifts etc… so that the timing of your one sec selection for analysis may not be neutral for AU145/AU47 occurence.
-Eye blinks may occur during close interactions as a consequence of menace reflex due to closeness of the interacting horse’s heads/body. In your data, except alert position, you recorded AU145 above chance in context behaviors including a physical contact (contact, nip, grasp and push) so that menace reflex could not be excluded in these contexts.
-In your “alert context behaviour”, even if you coded the behaviour according to well recognized alert positions, the horses are within one horse length of each other and so in close proximity: was their attention oriented toward the close horse or toward an external distant target? Having the head in alert high position help the horse to improve its visual acuity of targets located at some distance. I think it could be interesting to try to dissociate these two situations for your discussion. Indeed blinking has also been linked to social functions (e.g. in humans, in comparing SEB rate between looking at a still target, reading, and conversation, the highest blink rate was found in conversation). It could be sometime difficult to dissociate blinks related to attentional state from social or communicative functions. At my opinion, your experimental condition of close proximity may induce mixed influences on SEBR, attentional but also social that remains to be discovered.
L430: Merkies et al found a decrease in SEBR and increase in eyelid twitch
L432: Rashid at al. 2020 found an increase in AU47 as indicator of pain (not discussing positive emotion). Could you check your reference?
L 434: For modulation of SEBR during attentional behaviours in horses, see: "Spontaneous eye blinks in horses (Equus caballus) are modulated by attention | Scientific Reports,14, 19336 2024", https://doi.org/10.1038/s41598-024-70141-y.
L477-787: I would suggest to more in depth discussion about the attribution of AUH21 to the platysma muscle:
L477-478: AU21 is recorded in humans and gibbons as «neck tightener». It could be interesting to discuss/explain why you call it «face tightener» in horses.
L 479: The platysma has been identified in several species including chimpanzee, macaque, orang-outang, cat, dog for which a FACS has been developed but only described in FACS in humans and hylobatids. I would suggest an explanatory sentence as to why AU21 was not described in these species. This is important for a comparative aspect and could support the importance of your finding.
L481-482: In dogs, AU21 is not observed but the platysma is the muscle involved in the AU16 (lower lip depressor) and partly in the AU25 (lips part). I would suggest to integrate this in your discussion.
L480-487: Could you explain why you associate the facial appearance of prominence in this region to the contraction of the platysma muscle ? And why you are not including other chewing muscles of this region (like masseter e.g.)?

Experimental design

The experimental design should be better explained in Methods : Could you make it more clear what your experimental setup is ( L131-133) ? and it could be interesting to explain how the authors managed to have a high quality of image in field recordings which is very important for facial behaviour analysis (L146).

Validity of the findings

No comment

Additional comments

Thank you very much for the opportunity to review this manuscript. This new methodology to combine facial action units/action descriptors opens a fascinating area of research and is of high interest to gain comprehensive understanding of horse’s emotions, cognition and communicative actions.

This research was conducted to a high standard, with well-considered interpretations of the results.

I have some minor remarks, mainly to help improving reading and highlighting the points discussed (see basic reporting)

It seems that the legends of the figures and tables are lacking ? I couldn’t find them ?

Reviewer 5 ·

Basic reporting

o The manuscript is written in clear and professional English. A few suggestions for areas that could be revised for clarity:
In particular, the following two sentences in the abstract are quite long and would benefit from being split for readability: “Although the Equine Facial Action Coding System (EquiFACS) has characterised a wide range of equine (Equus caballus) facial movements (Action Units [AU] and Descriptors [AD]), there is still a lack of systematic documentation of whether and how these AUs and ADs are combined to produce discrete configurations of facial behaviour in horses….Facial behaviour was recorded during horse-horse interactions and a bank of 805 AU/AD combinations across 22 distinct behaviours occurring during affiliative (non-play), play, agonistic and attentional contexts was created using EquiFACS, based on the coding of contextual behaviour”.
o For improved clarity, please briefly describe the appearance of the newly defined movement in the abstract: “We also defined a new EquiFACS Action Unit, AUH21 (platysma), a movement found in humans and gibbons but no other species studied” – that looks how?
o Line 62: Please capitalize the 'C' in Canis familiaris.
Intro & background to show context.
o Line 75: The statement “There are also striking similarities in the facial behaviour of humans and other species” could benefit from acknowledging that there can be notable differences as well, such as those highlighted by Caeiro et al. (2017): Dogs and humans respond to emotionally competent stimuli by producing different facial actions (Scientific Reports, 7(1), 15525).
o Line 387ff: I suggest also including the following paper, which could not identify specific facial expressions associated with positive anticipation in horses: Ricci-Bonot, C., & Mills, D. S. (2023). Recognising the facial expression of frustration in the horse during feeding period. Applied Animal Behaviour Science, 265, 105966.
o Also, the following work is relevant for your paper: Hintze et al. (2016). Are eyes a mirror of the soul? What eye wrinkles reveal about a horse’s emotional state. PloS One, 11(10), e0164017.
o It would be helpful to clearly state in the introduction that facial behaviour comprises a variable number of AU/ADs shown in combination as this information can avoid confusion in the following sections.
o As suggested above, consider adding Caeiro et al. 2017 on differences in emotional facial expressions between species, Ricci-Bonot and Mills (2023) on recognizing positive anticipation and frustration in horses during feeding, and Hintze et al. (2016) on eye wrinkles as indicators of emotional state in horses, as these studies are relevant to the current work.
o Ensure consistency in formatting the in-text references. In section L56-66, in some cases, references appear in brackets within another bracket, as in "(Otaria flavescens, (Llamazares-Martín et al., 2017)" while in others, no brackets are used: (canis familiaris, Maglieri et al., 2022).
o For Table 4 and subsequent tables, consider visually highlighting the listed AU/ADs in the images—using arrows or markers—to clearly indicate the activated AU/ADs visible in each picture for the reader.

Experimental design

o This study is original research and relevant to the journal’s scope.
o The research question and objectives appear well-defined and relevant. The study addresses a recognized gap in equine facial behaviour research.
o Overall, the study meets high technical and ethical standards, though the reliability assessments and some methodological details need further clarification.
• Methods that are not provided with sufficient detail & information to replicate:
o Line 147: Provide more detail on how “social inactivity” is defined, also clarify why you chose three seconds as a threshold as this may be very brief for this definition of inactivity. Additionally, clarify the criteria for defining a “prolonged interaction” (Line 148) and the criteria for deciding when to stop recording mid-interaction and when to restart again – was this with a pause in between?
o Please specify the approximate distance between the horses and the camera used for observations.
o How many hours of video material were collected in total? This information would provide additional context for replication.
o Table 1: Since the ethogram is merged from three existing social ethograms, I recommend including the respective references for each behaviour variable listed, so they can be traced back to their sources.
o Line 155ff: More detail is needed on how behavioural contexts were determined using the ethogram in Tab. 1. For example, was observing a single rigid stance sufficient to classify a context as “alert – attentional”? How were ambiguous situations handled when behaviours from multiple contexts were present? Since this classification is central to your study, more detail on the classification process and especially an inter-coder reliability assessment is highly recommended.
o L164ff: The approach mentions balancing behaviours by date, time of day, and conspecific identity, followed by random selection in cases with multiple behaviours, but more details on this process would be helpful for reproducibility. Als,o the rationale for selecting five facial behaviours per context per horse might be based on practical or statistical considerations; however, including this reasoning—whether from past studies, pilot analyses, or empirical choices—would support the validity of this threshold.
o Line 175: If you selected behaviours based on peak intensity, could this approach overlook behaviours that are brief but significant? Clarify how this approach accounts for short-lived behaviours. A facial behaviour can consist of several AU/ADs and I assume that not all have their peak intensity at the same time – How did you handle this?
o Reliability assessment: In L191ff, the manuscript notes that 5% of videos from a larger dataset were coded by a second observer for reliability assessment. However, since the dataset used in this study is only a subset of that broader dataset, the actual percentage of videos coded by a second observer specifically for this study is likely even lower. Could you clarify what percentage of the videos used in this study were coded a second time? I strongly recommend conducting a more extensive reliability assessment with a higher percentage of videos coded by a second observer, as even 5%—and likely less for this subset—cannot be considered sufficient for a robust intercoder reliability assessment.
Also, Cohen’s Kappa is commonly used for binary data in intercoder reliability assessments. So please provide reasoning for the different method used.
Additionally, as described earlier, a reliability assessment for the selection of behavioural context coding would strengthen the study, as this seems to be absent but behavioural context is highly important to the study objectives.

Validity of the findings

o The study is novel and adds valuable insights into equine facial behaviour research. A more extensive reliability assessment could reinforce its impact.
o The manuscript emphasizes the study’s contribution to equine behaviour literature, supporting meaningful replication.
o The authors appear to provide the relevant data on the OSF Platform.
o Conclusions are appropriately linked to the research questions, though reliability measures, particularly for behavioural context classification, would strengthen the findings.

Additional comments

Table 4 and all subsequent tables: To enhance clarity, consider adding visual markers (e.g., arrows) on the images in these tables to indicate specific AUs/ADs.

---

## Round 0.2 · accepted · Accept

The authors have introduced all the suggestions provided by the reviewers. There is only one minor change that needs to be done for the manuscript to be ready for publication: "increased blinking has also been shown to be a marker of positive emotion in horses (Rashid et al., 2020)." This sentence needs to be corrected since in Rashid et al., the authors state that "increased frequency of half blink (AU47) as a new indicator of pain in the horses of this study.". And for blink (AU145) they do not make any associations with positive emotions.
With this change the manuscript will be ready for publication.

Line 520-521.

Reviewer 2 ·

Basic reporting

no comment

Experimental design

no comment

Validity of the findings

no comment

Additional comments

The manuscript was already well-written, clear, and rich with reference figures in its first submission. After this detailed and thorough revision by the authors, I have no further modifications to request.

·

Basic reporting

Very good

Experimental design

Excellent

Validity of the findings

Excellent

Additional comments

No further comments from me, the ms is a really nice piece of work

·

Basic reporting

Thank you for addressing all the comments so thoughtfully and completely. I would suggest very minor editing and after that is addressed, I recommend publication.

Line 517 : For the clarity of the text, I would suggest that you remove the reference of Merkies et al. but add line 520, that they observed together with an decrease in SEBR an increase in eyelid twitch (just a suggestion so that the reader would not be confused by the same reference after a sentence indicating an increase in SEBR and a second sentence indicating a decrease in SEBR).

Line 520-521 : “Rashid at al. 2020 found an increase in AU47 as indicator of pain (not discussing positive emotion). Could you check your reference? "
You indicated having removed this sentence but it seems that it still remains? Maybe something went wrong when editing ?

Experimental design

-

Validity of the findings

-

Additional comments

-